# Optimal Learning for Multi-pass Stochastic Gradient Methods

**Junhong Lin**
LCSL, IIT-MIT, USA
junhong.lin@iit.it

**Lorenzo Rosasco**
DIBRIS, Univ. Genova, ITALY
LCSL, IIT-MIT, USA
lrosasco@mit.edu

## Abstract

We analyze the learning properties of the stochastic gradient method when multiple passes over the data and mini-batches are allowed. In particular, we consider the square loss and show that for a universal step-size choice, the number of passes acts as a regularization parameter, and optimal finite sample bounds can be achieved by early-stopping. Moreover, we show that larger step-sizes are allowed when considering mini-batches. Our analysis is based on a unifying approach, encompassing both batch and stochastic gradient methods as special cases.

## 1 Introduction

Modern machine learning applications require computational approaches that are at the same time statistically accurate and numerically efficient [2]. This has motivated a recent interest in stochastic gradient methods (SGM), since on the one hand they enjoy good practical performances, especially in large scale scenarios, and on the other hand they are amenable to theoretical studies. In particular, unlike other learning approaches, such as empirical risk minimization or Tikhonov regularization, theoretical results on SGM naturally integrate statistical and computational aspects.

Most generalization studies on SGM consider the case where only one pass over the data is allowed and the step-size is appropriately chosen, [5, 14, 29, 26, 9, 16] (possibly considering averaging [18]). In particular, recent works show how the step-size can be seen to play the role of a regularization parameter whose choice controls the bias and variance properties of the obtained solution [29, 26, 9]. These latter works show that balancing these contributions, it is possible to derive a step-size choice leading to optimal learning bounds. Such a choice typically depends on some unknown properties of the data generating distributions and in practice can be chosen by cross-validation.

While processing each data point only once is natural in streaming/online scenarios, in practice SGM is often used as a tool for processing large data-sets and multiple passes over the data are typically considered. In this case, the number of passes over the data, as well as the step-size, need then to be determined. While the role of multiple passes is well understood if the goal is empirical risk minimization [3], its effect with respect to generalization is less clear and a few recent works have recently started to tackle this question. In particular, results in this direction have been derived in [10] and [11]. The former work considers a general stochastic optimization setting and studies stability properties of SGM allowing to derive convergence results as well as finite sample bounds. The latter work, restricted to supervised learning, further develops these results to compare the respective roles of step-size and number of passes, and show how different parameter settings can lead to optimal error bounds. In particular, it shows that there are two extreme cases: one between the step-size or the number of passes is fixed a priori, while the other one acts as a regularization parameter and needs to be chosen adaptively. The main shortcoming of these latter results is that they are in the worst case, in the sense that they do not consider the possible effect of capacity assumptions [30, 4] shown to lead to faster rates for other learning approaches such as Tikhonov regularization. Further, these results do not consider the possible effect of mini-batches, rather than a single point in each gradient

step [21, 8, 24, 15]. This latter strategy is often considered especially for parallel implementation of SGM.

The study in this paper, fills in these gaps in the case where the loss function is the least squares loss. We consider a variant of SGM for least squares, where gradients are sampled uniformly at random and mini-batches are allowed. The number of passes, the step-size and the mini-batch size are then parameters to be determined. Our main results highlight the respective roles of these parameters and show how can they be chosen so that the corresponding solutions achieve optimal learning errors. In particular, we show for the first time that multi-pass SGM with early stopping and a universal step-size choice can achieve optimal learning rates, matching those of ridge regression [23, 4]. Further, our analysis shows how the mini-batch size and the step-size choice are tightly related. Indeed, larger mini-batch sizes allow to consider larger step-sizes while keeping the optimal learning bounds. This result could give an insight on how to exploit mini-batches for parallel computations while preserving optimal statistical accuracy. Finally we note that a recent work [19] is tightly related to the analysis in the paper. The generalization properties of a multi-pass incremental gradient are analyzed in [19], for a cyclic, rather than a stochastic, choice of the gradients and with no mini-batches. The analysis in this latter case appears to be harder and results in [19] give good learning bounds only in restricted setting and considering iterates rather than the excess risk. Compared to [19] our results show how stochasticity can be exploited to get faster capacity dependent rates and analyze the role of mini-batches. The basic idea of our proof is to approximate the SGM learning sequence in terms of the batch GM sequence, see Subsection 3.4 for further details. This thus allows one to study batch and stochastic gradient methods simultaneously, and may be also useful for analysing other learning algorithms.

The rest of this paper is organized as follows. Section 2 introduces the learning setting and the SGM algorithm. Main results with discussions and proof sketches are presented in Section 3. Finally, simple numerical simulations are given in Section 4 to complement our theoretical results.

**Notation** For any $a, b \in \mathbb{R}$, $a \vee b$ denotes the maximum of $a$ and $b$. $\mathbb{N}$ is the set of all positive integers. For any $T \in \mathbb{N}$, $[T]$ denotes the set $\{1, \cdots, T\}$. For any two positive sequences $\{a_t\}_{t \in [T]}$ and $\{b_t\}_{t \in [T]}$, the notation $a_t \lesssim b_t$ for all $t \in [T]$ means that there exists a positive constant $C \geq 0$ such that $C$ is independent of $t$ and that $a_t \leq C b_t$ for all $t \in [T]$.

## 2 Learning with SGM

We begin by introducing the learning setting we consider, and then describe the SGM learning algorithm. Following [19], the formulation we consider is close to the setting of functional regression, and covers the reproducing kernel Hilbert space (RKHS) setting as a special case. In particular, it reduces to standard linear regression for finite dimensions.

### 2.1 Learning Problems

Let $H$ be a separable Hilbert space, with inner product and induced norm denoted by $\langle \cdot, \cdot \rangle_H$ and $\| \cdot \|_H$, respectively. Let the input space $X \subseteq H$ and the output space $Y \subseteq \mathbb{R}$. Let $\rho$ be an unknown probability measure on $Z = X \times Y$, $\rho_X(\cdot)$ the induced marginal measure on $X$, and $\rho(\cdot|x)$ the conditional probability measure on $Y$ with respect to $x \in X$ and $\rho$.

Considering the square loss function, the problem under study is the minimization of the *risk*,

$$\inf_{\omega \in H} \mathcal{E}(\omega), \quad \mathcal{E}(\omega) = \int_{X \times Y} (\langle \omega, x \rangle_H - y)^2 d\rho(x, y), \tag{1}$$

when the measure $\rho$ is known only through a sample $\mathbf{z} = \{z_i = (x_i, y_i)\}_{i=1}^{m}$ of size $m \in \mathbb{N}$, independently and identically distributed (i.i.d.) according to $\rho$. In the following, we measure the quality of an approximate solution $\hat{\omega} \in H$ (an estimator) considering *the excess risk*, i.e.,

$$\mathcal{E}(\hat{\omega}) - \inf_{\omega \in H} \mathcal{E}(\omega). \tag{2}$$

Throughout this paper, we assume that there exists a constant $\kappa \in [1, \infty[$, such that

$$\langle x, x' \rangle_H \leq \kappa^2, \quad \forall x, x' \in X. \tag{3}$$

### 2.2 Stochastic Gradient Method

We study the following SGM (with mini-batches, without penalization or constraints).

**Algorithm 1.** *Let $b \in [m]$. Given any sample* **z**, *the b-minibatch stochastic gradient method is defined by $\omega_1 = 0$ and*

$$\omega_{t+1} = \omega_t - \eta_t \frac{1}{b} \sum_{i=b(t-1)+1}^{bt} (\langle \omega_t, x_{j_i} \rangle_H - y_{j_i}) x_{j_i}, \qquad t = 1, \ldots, T, \tag{4}$$

*where $\{\eta_t > 0\}$ is a step-size sequence. Here, $j_1, j_2, \cdots, j_{bT}$ are independent and identically distributed (i.i.d.) random variables from the uniform distribution on $[m]$* [1].

Different choices for the (mini-)batch size $b$ can lead to different algorithms. In particular, for $b = 1$, the above algorithm corresponds to a simple SGM, while for $b = m$, it is a stochastic version of the batch gradient descent.

The aim of this paper is to derive excess risk bounds for the above algorithm under appropriate assumptions. Throughout this paper, we assume that $\{\eta_t\}_t$ is non-increasing, and $T \in \mathbb{N}$ with $T \geq 3$. We denote by $\mathbf{J}_t$ the set $\{j_l : l = b(t-1) + 1, \cdots, bt\}$ and by $\mathbf{J}$ the set $\{j_l : l = 1, \cdots, bT\}$.

## 3 Main Results with Discussions

In this section, we first state some basic assumptions. Then, we present and discuss our main results.

### 3.1 Assumptions

The following assumption is related to a moment hypothesis on $|y|^2$. It is weaker than the often considered bounded output assumption, and trivially verified in binary classification problems where $Y = \{-1, 1\}$.

**Assumption 1.** *There exists constants $M \in ]0, \infty[$ and $v \in ]1, \infty[$ such that*

$$\int_Y y^{2l} d\rho(y|x) \leq l! M^l v, \quad \forall l \in \mathbb{N}, \tag{5}$$

$\rho_X$-*almost surely.*

To present our next assumption, we introduce the operator $\mathcal{L} : L^2(H, \rho_X) \to L^2(H, \rho_X)$, defined by $\mathcal{L}(f) = \int_X \langle x, \cdot \rangle_H f(x) \rho_X(x)$. Under Assumption (3), $\mathcal{L}$ can be proved to be positive trace class operators, and hence $\mathcal{L}^\zeta$ with $\zeta \in \mathbb{R}$ can be defined by using the spectrum theory [7].

The Hilbert space of square integral functions from $H$ to $\mathbb{R}$ with respect to $\rho_X$, with induced norm given by $\|f\|_\rho = \left( \int_X |f(x)|^2 d\rho_X(x) \right)^{1/2}$, is denoted by $(L^2(H, \rho_X), \| \cdot \|_\rho)$. It is well known that the function minimizing $\int_Z (f(x) - y)^2 d\rho(z)$ over all measurable functions $f : H \to \mathbb{R}$ is the regression function, which is given by

$$f_\rho(x) = \int_Y y d\rho(y|x), \qquad x \in X. \tag{6}$$

Define another Hilbert space $H_\rho = \{f : X \to \mathbb{R} | \exists \omega \in H \text{ with } f(x) = \langle \omega, x \rangle_H, \rho_X\text{-almost surely}\}$. Under Assumption 3, it is easy to see that $H_\rho$ is a subspace of $L^2(H, \rho_X)$. Let $f_{\mathcal{H}}$ be the projection of the regression function $f_\rho$ onto the closure of $H_\rho$ in $L^2(H, \rho_X)$. It is easy to see that the search for a solution of Problem (1) is equivalent to the search of a linear function from $H_\rho$ to approximate $f_{\mathcal{H}}$. From this point of view, bounds on the excess risk of a learning algorithm naturally depend on the following assumption, which quantifies how well, the target function $f_{\mathcal{H}}$ can be approximated by $H_\rho$.

**Assumption 2.** *There exist $\zeta > 0$ and $R > 0$, such that $\|\mathcal{L}^{-\zeta} f_{\mathcal{H}}\|_\rho \leq R$.*

The above assumption is fairly standard [7, 19] in non-parametric regression. The bigger $\zeta$ is, the more stringent the assumption is, since $\mathcal{L}^{\zeta_1}(L^2(H, \rho_X)) \subseteq \mathcal{L}^{\zeta_2}(L^2(H, \rho_X))$ when $\zeta_1 \geq \zeta_2$. In particular, for $\zeta = 0$, we are assuming $\|f_{\mathcal{H}}\|_\rho < \infty$, while for $\zeta = 1/2$, we are requiring $f_{\mathcal{H}} \in H_\rho$, since [25, 19]

$$H_\rho = \mathcal{L}^{1/2}(L^2(H, \rho_X)).$$

Finally, the last assumption relates to the capacity of the hypothesis space.

**Assumption 3.** *For some $\gamma \in ]0,1]$ and $c_\gamma > 0$, $\mathcal{L}$ satisfies*

$$\mathrm{tr}(\mathcal{L}(\mathcal{L} + \lambda I)^{-1}) \leq c_\gamma \lambda^{-\gamma}, \quad \text{for all } \lambda > 0. \tag{7}$$

The LHS of (7) is called as the effective dimension, or the degrees of freedom [30, 4]. It can be related to covering/entropy number conditions, see [25] for further details. Assumption 3 is always true for $\gamma = 1$ and $c_\gamma = \kappa^2$, since $\mathcal{L}$ is a trace class operator which implies the eigenvalues of $\mathcal{L}$, denoted as $\sigma_i$, satisfy $\mathrm{tr}(\mathcal{L}) = \sum_i \sigma_i \leq \kappa^2$. This is referred as the capacity independent setting. Assumption 3 with $\gamma \in ]0,1]$ allows to derive better error rates. It is satisfied, e.g., if the eigenvalues of $\mathcal{L}$ satisfy a polynomial decaying condition $\sigma_i \sim i^{-1/\gamma}$, or with $\gamma = 0$ if $\mathcal{L}$ is finite rank.

## 3.2 Main Results

We start with the following corollary, which is a simplified version of our main results stated next.

**Corollary 3.1.** *Under Assumptions 2 and 3, let $\zeta \geq 1/2$ and $|y| \leq M$ $\rho_X$-almost surely for some $M > 0$. Consider the SGM with*
*1) $p_* = \lceil m^{\frac{1}{2\zeta+\gamma}} \rceil$, $b = 1$, $\eta_t \simeq \frac{1}{m}$ for all $t \in [(p_* m)]$, and $\tilde{\omega}_{p_*} = \omega_{p_* m+1}$.*
*If $m$ is large enough, with high probability[2], there holds*

$$\mathbb{E}_{\mathbf{J}}[\mathcal{E}(\tilde{\omega}_{p_*})] - \inf_{\omega \in H} \mathcal{E} \lesssim m^{-\frac{2\zeta}{2\zeta+\gamma}}.$$

*Furthermore, the above also holds for the SGM with[3]*
*2) or $p_* = \lceil m^{\frac{1}{2\zeta+\gamma}} \rceil$, $b = \sqrt{m}$, $\eta_t \simeq \frac{1}{\sqrt{m}}$ for all $t \in [(p_* \sqrt{m})]$, and $\tilde{\omega}_{p_*} = \omega_{p_* \sqrt{m}+1}$.*

In the above, $p_*$ is the number of 'passes' over the data, which is defined as $\lceil \frac{bt}{m} \rceil$ at $t$ iterations. The above result asserts that, at $p_*$ passes over the data, the simple SGM with fixed step-size achieves optimal learning error bounds, matching those of ridge regression [4]. Furthermore, using mini-batch allows to use a larger step-size while achieving the same optimal error bounds.

**Remark 3.2** (Finite Dimensional Case). *With a simple modification of our proofs, we can derive similar results for the finite dimensional case, i.e., $H = \mathbb{R}^d$, where in this case, $\gamma = 0$. In particular, letting $\zeta = 1/2$, under the same assumptions of Corollary 3.1, if one considers the SGM with $b = 1$ and $\eta_t \simeq \frac{1}{m}$ for all $t \in m^2$, then with high probability, $\mathbb{E}_{\mathbf{J}}[\mathcal{E}(\omega_{m^2+1})] - \inf_{\omega \in H} \mathcal{E} \lesssim d/m$, provided that $m \gtrsim d \log d$.*

Our main theorem of this paper is stated next, and provides error bounds for the studied algorithm. For the sake of readability, we only consider the case $\zeta \geq 1/2$ in a fixed step-size setting. General results in a more general setting ($\eta_t = \eta_1 t^{-\theta}$ with $0 \leq \theta < 1$, and/or the case $\zeta \in ]0,1/2]$) can be found in the appendix.

**Theorem 3.3.** *Under Assumptions 1, 2 and 3, let $\zeta \geq 1/2$, $\delta \in ]0,1[$, $\eta_t = \eta \kappa^{-2}$ for all $t \in [T]$, with $\eta \leq \frac{1}{8(\log T+1)}$. If $m \geq m_\delta$, then the following holds with probability at least $1 - \delta$: for all $t \in [T]$,*

$$\mathbb{E}_{\mathbf{J}}[\mathcal{E}(\omega_{t+1})] - \inf_{\omega \in H} \mathcal{E} \leq q_1 (\eta t)^{-2\zeta} + q_2 m^{-\frac{2\zeta}{2\zeta+\gamma}} (1 + m^{-\frac{1}{2\zeta+\gamma}} \eta t)^2 \log^2 T \log^2 \frac{1}{\delta} \tag{8}$$
$$+ q_3 \eta b^{-1} (1 \vee m^{-\frac{1}{2\zeta+\gamma}} \eta t) \log T.$$

*Here, $m_\delta, q_1, q_2$ and $q_3$ are positive constants depending on $\kappa^2, \|\mathcal{T}\|, M, v, \zeta, R, c_\gamma, \gamma$, and $m_\delta$ also on $\delta$ (which will be given explicitly in the proof).*

There are three terms in the upper bounds of (8). The first term depends on the regularity of the target function and it arises from bounding the bias, while the last two terms result from estimating the sample variance and the computational variance (due to the random choices of the points), respectively. To derive optimal rates, it is necessary to balance these three terms. Solving this trade-off problem leads to different choices on $\eta$, $T$, and $b$, corresponding to different regularization strategies, as shown in subsequent corollaries.

The first corollary gives generalization error bounds for SGM, with a universal step-size depending on the number of sample points.

**Corollary 3.4.** *Under Assumptions 1, 2 and 3, let $\zeta \geq 1/2$, $\delta \in ]0,1[$, $b = 1$ and $\eta_t \simeq \frac{1}{m}$ for all $t \in [T]$, where $T \leq m^2$. If $m \geq m_\delta$, then with probability at least $1 - \delta$, there holds*

$$\mathbb{E}_{\mathbf{J}}[\mathcal{E}(\omega_{t+1})] - \inf_{\omega \in H} \mathcal{E} \lesssim \left\{ \left(\frac{m}{t}\right)^{2\zeta} + m^{-\frac{2\zeta+2}{2\zeta+\gamma}} \left(\frac{t}{m}\right)^2 \right\} \cdot \log^2 m \log^2 \frac{1}{\delta}, \quad \forall t \in [T], \quad (9)$$

*and in particular,*

$$\mathbb{E}_{\mathbf{J}}[\mathcal{E}(\omega_{T^*+1})] - \inf_{\omega \in H} \mathcal{E} \lesssim m^{-\frac{2\zeta}{2\zeta+\gamma}} \log^2 m \log^2 \frac{1}{\delta}, \quad (10)$$

*where $T^* = \lceil m^{\frac{2\zeta+\gamma+1}{2\zeta+\gamma}} \rceil$. Here, $m_\delta$ is exactly the same as in Theorem 3.3.*

**Remark 3.5.** *Ignoring the logarithmic term and letting $t = pm$, Eq. (9) becomes*

$$\mathbb{E}_{\mathbf{J}}[\mathcal{E}(\omega_{pm+1})] - \inf_{\omega \in H} \mathcal{E} \lesssim p^{-2\zeta} + m^{-\frac{2\zeta+2}{2\zeta+\gamma}} p^2.$$

*A smaller $p$ may lead to a larger bias, while a larger $p$ may lead to a larger sample error. From this point of view, $p$ has a regularization effect.*

The second corollary provides error bounds for SGM with a fixed mini-batch size and a fixed step-size (which depend on the number of sample points).

**Corollary 3.6.** *Under Assumptions 1, 2 and 3, let $\zeta \geq 1/2$, $\delta \in ]0,1[$, $b = \lceil \sqrt{m} \rceil$ and $\eta_t \simeq \frac{1}{\sqrt{m}}$ for all $t \in [T]$, where $T \leq m^2$. If $m \geq m_\delta$, then with probability at least $1 - \delta$, there holds*

$$\mathbb{E}_{\mathbf{J}}[\mathcal{E}(\omega_{t+1})] - \inf_{\omega \in H} \mathcal{E} \lesssim \left\{ \left(\frac{\sqrt{m}}{t}\right)^{2\zeta} + m^{-\frac{2\zeta+2}{2\zeta+\gamma}} \left(\frac{t}{\sqrt{m}}\right)^2 \right\} \log^2 m \log^2 \frac{1}{\delta}, \quad \forall t \in [T], \quad (11)$$

*and particularly,*

$$\mathbb{E}_{\mathbf{J}}[\mathcal{E}(\omega_{T^*+1})] - \inf_{\omega \in H} \mathcal{E} \lesssim m^{-\frac{2\zeta}{2\zeta+\gamma}} \log^2 m \log^2 \frac{1}{\delta}, \quad (12)$$

*where $T^* = \lceil m^{\frac{1}{2\zeta+\gamma} + \frac{1}{2}} \rceil$.*

The above two corollaries follow from Theorem 3.3 with the simple observation that the dominating terms in (8) are the terms related to the bias and the sample variance, when a small step-size is chosen. The only free parameter in (9) and (11) is the number of iterations/passes.

The ideal stopping rule is achieved by balancing the two terms related to the bias and the sample variance, showing the regularization effect of the number of passes. Since the ideal stopping rule depends on the unknown parameters $\zeta$ and $\gamma$, a hold-out cross-validation procedure is often used to tune the stopping rule in practice. Using an argument similar to that in Chapter 6 from [25], it is possible to show that this procedure can achieve the same convergence rate.

We give some further remarks. First, the upper bound in (10) is optimal up to a logarithmic factor, in the sense that it matches the minimax lower rate in [4]. Second, according to Corollaries 3.4 and 3.6, $\frac{bT^*}{m} \simeq m^{\frac{1}{2\zeta+\gamma}}$ passes over the data are needed to obtain optimal rates in both cases. Finally, in comparing the simple SGM and the mini-batch SGM, Corollaries 3.4 and 3.6 show that a larger step-size is allowed to use for the latter.

In the next result, both the step-size and the stopping rule are tuned to obtain optimal rates for simple SGM with multiple passes. In this case, the step-size and the number of iterations are the regularization parameters.

**Corollary 3.7.** *Under Assumptions 1, 2 and 3, let $\zeta \geq 1/2$, $\delta \in ]0,1[$, $b = 1$ and $\eta_t \simeq m^{-\frac{2\zeta}{2\zeta+\gamma}}$ for all $t \in [T]$, where $T \leq m^2$. If $m \geq m_\delta$, and $T^* = \lceil m^{\frac{2\zeta+1}{2\zeta+\gamma}} \rceil$, then (10) holds with probability at least $1 - \delta$.*

**Remark 3.8.** *If we make no assumption on the capacity, i.e., $\gamma = 1$, Corollary 3.7 recovers the result in [29] for one pass SGM.*

The next corollary shows that for some suitable mini-batch sizes, optimal rates can be achieved with a constant step-size (which is nearly independent of the number of sample points) by early stopping.

**Corollary 3.9.** *Under Assumptions 1, 2 and 3, let $\zeta \geq 1/2$, $\delta \in ]0,1[$, $b = \lceil m^{\frac{2\zeta}{2\zeta+\gamma}} \rceil$ and $\eta_t \simeq \frac{1}{\log m}$ for all $t \in [T]$, where $T \leq m^2$. If $m \geq m_\delta$, and $T^* = \lceil m^{\frac{1}{2\zeta+\gamma}} \rceil$, then (10) holds with probability at least $1 - \delta$.*

According to Corollaries 3.7 and 3.9, around $m^{\frac{1-\gamma}{2\zeta+\gamma}}$ passes over the data are needed to achieve the best performance in the above two strategies. In comparisons with Corollaries 3.4 and 3.6 where around $m^{\frac{\zeta+1}{2\zeta+\gamma}}$ passes are required, the latter seems to require fewer passes over the data. However, in this case, one might have to run the algorithms multiple times to tune the step-size, or the mini-batch size.

Finally, the last result gives generalization error bounds for 'batch' SGM with a constant step-size (nearly independent of the number of sample points).

**Corollary 3.10.** *Under Assumptions 1, 2 and 3, let $\zeta \geq 1/2$, $\delta \in ]0, 1[$, $b = m$ and $\eta_t \simeq \frac{1}{\log m}$ for all $t \in [T]$, where $T \leq m^2$. If $m \geq m_\delta$, and $T^* = \lceil m^{\frac{1}{2\zeta+\gamma}} \rceil$, then (10) holds with probability at least $1 - \delta$.*

As will be seen in the proof from the appendix, the above result also holds when replacing the sequence $\{\omega_t\}$ by the sequence $\{\nu_t\}_t$ generated from batch GM in (14). In this sense, we study the gradient-based learning algorithms simultaneously.

### 3.3 Discussions

We compare our results with previous works. For non-parametric regression with the square loss, one pass SGM has been studied in, e.g., [29, 22, 26, 9]. In particular, [29] proved capacity independent rate of order $O(m^{-\frac{2\zeta}{2\zeta+1}} \log m)$ with a fixed step-size $\eta \simeq m^{-\frac{2\zeta}{2\zeta+1}}$, and [9] derived capacity dependent error bounds of order $O(m^{-\frac{2\min(\zeta,1)}{2\min(\zeta,1)+\gamma}})$ (when $2\zeta + \gamma > 1$) for the average. Note also that a regularized version of SGM has been studied in [26], where the derived convergence rate there is of order $O(m^{-\frac{2\zeta}{2\zeta+1}})$ assuming that $\zeta \in [\frac{1}{2}, 1]$. In comparison with these existing convergence rates, our rates from (10) are comparable, either involving the capacity condition, or allowing a broader regularity parameter $\zeta$ (which thus improves the rates).

More recently, [19] studied multiple passes SGM with a fixed ordering at each pass, also called incremental gradient method. Making no assumption on the capacity, rates of order $O(m^{-\frac{\zeta}{\zeta+1}})$ (in $L^2(H, \rho_X)$-norm) with a universal step-size $\eta \simeq 1/m$ are derived. In comparisons, Corollary 3.4 achieves better rates, while considering the capacity assumption. Note also that [19] proved sharp rate in $H$-norm for $\zeta \geq 1/2$ in the capacity independent case. In fact, we can extend our analysis to the $H$-norm for Algorithm 4. We postpone this extension to a longer version of this paper.

The idea of using mini-batches (and parallel implements) to speed up SGM in a general stochastic optimization setting can be found, e.g., in [21, 8, 24, 15]. Our theoretical findings, especially the interplay between the mini-batch size and the step-size, can give further insights on parallelization learning. Besides, it has been shown in [6, 8] that for one pass mini-batch SGM with a fixed step-size $\eta \simeq b/\sqrt{m}$ and a smooth loss function, assuming the existence of at least one solution in the hypothesis space for the expected risk minimization, the convergence rate is of order $O(\sqrt{1/m} + b/m)$ by considering an averaging scheme. When adapting to the learning setting we consider, this reads as that if $f_\mathcal{H} \in H_\rho$, i.e., $\zeta = 1/2$, the convergence rate for the average is $O(\sqrt{1/m} + b/m)$. Note that, $f_\mathcal{H}$ does not necessarily belong to $H_\rho$ in general. Also, our derived convergence rate from Corollary 3.6 is better, when the regularity parameter $\zeta$ is greater than $1/2$, or $\gamma$ is smaller than 1.

### 3.4 Proof Sketch (Error Decomposition)

The key to our proof is a novel error decomposition, which may be also used in analysing other learning algorithms. One may also use the approach in [12, 11] which is based on the error decomposition, i.e., for some suitably intermediate element $\tilde{\omega} \in H$,

$$\mathbb{E}\mathcal{E}(\omega_t) - \inf_{w \in H} \mathcal{E} = [\mathbb{E}(\mathcal{E}(\omega_t) - \mathcal{E}_\mathbf{z}(\omega_t)) + \mathbb{E}\mathcal{E}_\mathbf{z}(\tilde{\omega}) - \mathcal{E}(\tilde{\omega})] + \mathbb{E}(\mathcal{E}_\mathbf{z}(\omega_t) - \mathcal{E}_\mathbf{z}(\tilde{\omega})) + \mathcal{E}(\tilde{\omega}) - \inf_{\omega \in H} \mathcal{E},$$

where $\mathcal{E}_\mathbf{z}$ denotes the empirical risk. However, one can only derive a sub-optimal convergence rate, since the proof procedure involves upper bounding the learning sequence to estimate the sample error (the first term of RHS). In this case the 'regularity' of the regression function can not be fully adapted for bounding the bias (the last term). Thanks to the property of squares loss, we can exploit a different error decomposition leading to better results.

We first introduce two sequences. The *population iteration* is defined by $\mu_1 = 0$ and

$$\mu_{t+1} = \mu_t - \eta_t \int_X (\langle \mu_t, x \rangle_H - f_\rho(x)) x \, d\rho_X(x), \qquad t = 1, \ldots, T. \tag{13}$$

The above iterated procedure is ideal and can not be implemented in practice, since the distribution $\rho_X$ is unknown in general. Replacing $\rho_X$ by the empirical measure and $f_\rho(x_i)$ by $y_i$, we derive the *sample iteration* (associated with the sample **z**), i.e., $\nu_1 = 0$ and

$$\nu_{t+1} = \nu_t - \eta_t \frac{1}{m} \sum_{i=1}^{m} (\langle \nu_t, x_i \rangle_H - y_i) x_i, \qquad t = 1, \ldots, T. \tag{14}$$

Clearly, $\mu_t$ is deterministic and $\nu_t$ is a $H$-valued random variable depending on **z**. Given the sample **z**, the sequence $\{\nu_t\}_t$ has a natural relationship with the learning sequence $\{\omega_t\}_t$, since

$$\mathbb{E}_{\mathbf{J}}[\omega_t] = \nu_t. \tag{15}$$

Indeed, taking the expectation with respect to $\mathbf{J}_t$ on both sides of (4), and noting that $\omega_t$ depends only on $\mathbf{J}_1, \cdots, \mathbf{J}_{t-1}$ (given any **z**), one has $\mathbb{E}_{\mathbf{J}_t}[\omega_{t+1}] = \omega_t - \eta_t \frac{1}{m} \sum_{i=1}^{m} (\langle \omega_t, x_i \rangle_H - y_i) x_i$, and thus, $\mathbb{E}_{\mathbf{J}}[\omega_{t+1}] = \mathbb{E}_{\mathbf{J}}[\omega_t] - \eta_t \frac{1}{m} \sum_{i=1}^{m} (\langle \mathbb{E}_{\mathbf{J}}[\omega_t], x_i \rangle_H - y_i) x_i, t = 1, \ldots, T$, which satisfies the iterative relationship given in (14). By an induction argument, (15) can then be proved.

Let $\mathcal{S}_\rho : H \to L^2(H, \rho_X)$ be the linear map defined by $(\mathcal{S}_\rho \omega)(x) = \langle \omega, x \rangle_H, \forall \omega, x \in H$. We have the following error decomposition.

**Proposition 3.11.** *We have*

$$\mathbb{E}_{\mathbf{J}}[\mathcal{E}(\omega_t)] - \inf_{f \in H} \mathcal{E}(f) \le 2\|\mathcal{S}_\rho \mu_t - f_\mathcal{H}\|_\rho^2 + 2\|\mathcal{S}_\rho \nu_t - \mathcal{S}_\rho \mu_t\|_\rho^2 + \mathbb{E}_{\mathbf{J}}[\|\mathcal{S}_\rho \omega_t - \mathcal{S}_\rho \nu_t\|^2]. \tag{16}$$

*Proof.* For any $\omega \in H$, we have [25, 19] $\mathcal{E}(\omega) - \inf_{f \in H} \mathcal{E}(f) = \|\mathcal{S}_\rho \omega - f_\mathcal{H}\|_\rho^2$. Thus, $\mathcal{E}(\omega_t) - \inf_{f \in H} \mathcal{E}(f) = \|\mathcal{S}_\rho \omega_t - f_\mathcal{H}\|_\rho^2$, and

$$\mathbb{E}_{\mathbf{J}}[\|\mathcal{S}_\rho \omega_t - f_\mathcal{H}\|_\rho^2] = \mathbb{E}_{\mathbf{J}}[\|\mathcal{S}_\rho \omega_t - \mathcal{S}_\rho \nu_t + \mathcal{S}_\rho \nu_t - f_\mathcal{H}\|_\rho^2]$$
$$= \mathbb{E}_{\mathbf{J}}[\|\mathcal{S}_\rho \omega_t - \mathcal{S}_\rho \nu_t\|_\rho^2 + \|\mathcal{S}_\rho \nu_t - f_\mathcal{H}\|_\rho^2] + 2\mathbb{E}_{\mathbf{J}} \langle \mathcal{S}_\rho \omega_t - \mathcal{S}_\rho \nu_t, \mathcal{S}_\rho \nu_t - f_\mathcal{H} \rangle_\rho.$$

Using (15) to the above, we get $\mathbb{E}_{\mathbf{J}}[\|\mathcal{S}_\rho \omega_t - f_\mathcal{H}\|_\rho^2] = \mathbb{E}_{\mathbf{J}}[\|\mathcal{S}_\rho \omega_t - \mathcal{S}_\rho \nu_t\|_\rho^2 + \|\mathcal{S}_\rho \nu_t - f_\mathcal{H}\|_\rho^2]$. Now the proof can be finished by considering

$$\|\mathcal{S}_\rho \nu_t - f_\mathcal{H}\|_\rho^2 = \|\mathcal{S}_\rho \nu_t - \mathcal{S}_\rho \mu_t + \mathcal{S}_\rho \mu_t - f_\mathcal{H}\|_\rho^2 \le 2\|\mathcal{S}_\rho \nu_t - \mathcal{S}_\rho \mu_t\|_\rho^2 + 2\|\mathcal{S}_\rho \mu_t - \mathcal{S}_\rho f_\mathcal{H}\|_\rho^2.$$

$\square$

There are three terms in the upper bound of the error decomposition (16). We refer to the deterministic term $\|\mathcal{S}_\rho \mu_t - f_\mathcal{H}\|_\rho^2$ as the *bias*, the term $\|\mathcal{S}_\rho \nu_t - \mathcal{S}_\rho \mu_t\|_\rho^2$ depending on **z** as the *sample variance*, and $\mathbb{E}_{\mathbf{J}}[\|\mathcal{S}_\rho \omega_t - \mathcal{S}_\rho \nu_t\|_\rho^2]$ as the *computational variance*. The bias term is deterministic and is well studied in the literature, see e.g., [28] and also [19]. The main novelties are the estimate of the sample and computational variances. The proof of these results is quite lengthy and makes use of some ideas from [28, 23, 1, 29, 26, 20]. These three error terms will be estimated in the appendix, see Lemma B.2, Theorem C.6 and Theorem D.9. The bound in Theorem 3.3 thus follows plugging these estimations in the error decomposition.

## 4   Numerical Simulations

In order to illustrate our theoretical results and the error decomposition, we first performed some simulations on a simple problem. We constructed $m = 100$ i.i.d. training examples of the form $y = f_\rho(x_i) + \omega_i$. Here, the regression function is $f_\rho(x) = |x - 1/2| - 1/2$, the input point $x_i$ is uniformly distributed in $[0, 1]$, and $\omega_i$ is a Gaussian noise with zero mean and standard deviation 1, for each $i \in [m]$. We perform three experiments with the same $H$, a RKHS associated with a Gaussian kernel $K(x, x') = \exp(-(x - x')^2/(2\sigma^2))$ where $\sigma = 0.2$. In the first experiment, we run mini-batch SGM, where the mini-batch size $b = \sqrt{m}$, and the step-size $\eta_t = 1/(8\sqrt{m})$. In the second experiment, we run simple SGM where the step-size is fixed as $\eta_t = 1/(8m)$, while in the third experiment, we run batch GM using the fixed step-size $\eta_t = 1/8$. For mini-batch SGM and SGM, the total error $\|\mathcal{S}_\rho \omega_t - f_\rho\|_{L_{\hat{\rho}}^2}^2$, the bias $\|\mathcal{S}_\rho \hat{\mu}_t - f_\rho\|_{L_{\hat{\rho}}^2}^2$, the sample variance $\|\mathcal{S}_\rho \nu_t - \mathcal{S}_\rho \hat{\mu}_t\|_{L_{\hat{\rho}}^2}^2$ and the computational variance $\|\mathcal{S}_\rho \omega_t - \mathcal{S}_\rho \nu_t\|_{L_{\hat{\rho}}^2}^2$, averaged over 50 trials, are depicted in Figures 1a and 1b, respectively. For batch GM, the total error $\|\mathcal{S}_\rho \nu_t - f_\rho\|_{L_{\hat{\rho}}^2}^2$, the bias $\|\mathcal{S}_\rho \hat{\mu}_t - f_\rho\|_{L_{\hat{\rho}}^2}^2$ and the

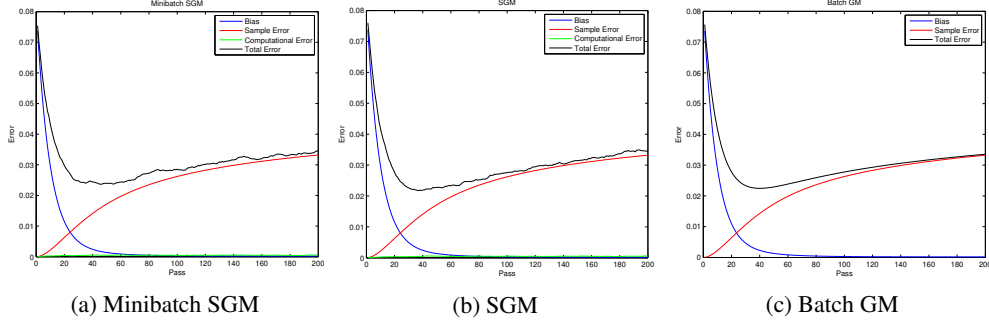

(a) Minibatch SGM        (b) SGM        (c) Batch GM

Figure 1: Error decompositions for gradient-based learning algorithms on *synthesis data*, where *m = 100*.

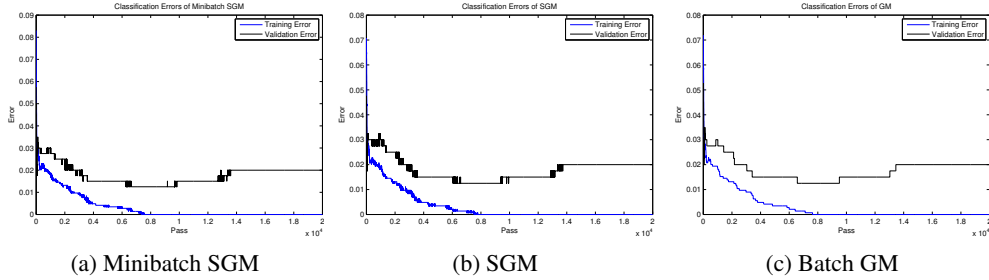

(a) Minibatch SGM        (b) SGM        (c) Batch GM

Figure 2: Misclassification Errors for gradient-based learning algorithms on *BreastCancer* dataset.

sample variance $\|\mathcal{S}_\rho \nu_t - \hat{\mu}_t\|^2_{L^2_{\hat{\rho}}}$, averaged over 50 trials are depicted in Figure 1c. Here, we replace the unknown marginal distribution $\rho_X$ by an empirical measure $\hat{\rho} = \frac{1}{2000} \sum_{i=1}^{2000} \delta_{\hat{x}_i}$, where each $\hat{x}_i$ is uniformly distributed in $[0,1]$. From Figure 1a or 1b, we see that as the number of passes increases[4], the bias decreases, while the sample error increases. Furthermore, we see that in comparisons with the bias and the sample error, the computational error is negligible. In all these experiments, the minimal total error is achieved when the bias and the sample error are balanced. These empirical results show the effects of the three terms from the error decomposition, and complement the derived bound (8), as well as the regularization effect of the number of passes over the data. Finally, we tested the simple SGM, mini-batch SGM, and batch GM, using similar step-sizes as those in the first simulation, on the *BreastCancer* data-set [5]. The classification errors on the training set and the testing set of these three algorithms are depicted in Figure 2. We see that all of these algorithms perform similarly, which complement the bounds in Corollaries 3.4, 3.6 and 3.10.

### Acknowledgments

This material is based upon work supported by the Center for Brains, Minds and Machines (CBMM), funded by NSF STC award CCF-1231216. L. R. acknowledges the financial support of the Italian Ministry of Education, University and Research FIRB project RBFR12M3AC.

## Footnotes

[1] Note that, the random variables $j_1, \cdots, j_{bT}$ are conditionally independent given the sample **z**.

[2]Here, 'high probability' refers to the sample $\mathbf{z}$.

[3]Here, we assume that $\sqrt{m}$ is an integer.

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
