[Supplementary Material · optMSGMsupplemental.pdf]

# Supplemental Materials for "Optimal Learning for Multi-pass Stochastic Gradient Methods"

## A  Preliminary

### A.1  Notation

We first introduce some notations. For $t \in \mathbb{N}$, $\Pi_{t+1}^T(L) = \prod_{k=t+1}^T (I - \eta_k L)$ for $t \in [T-1]$ and $\Pi_{T+1}^T(L) = I$, for any operator $L : \mathcal{H} \to \mathcal{H}$, where $\mathcal{H}$ is a Hilbert space and $I$ denotes the identity operator on $\mathcal{H}$. $\mathbb{E}[\xi]$ denotes the expectation of a random variable $\xi$. For a given bounded operator $L : L^2(H, \rho_X) \to H$, $\|L\|$ denotes the operator norm of $L$, i.e., $\|L\| = \sup_{f \in L^2(H, \rho_X), \|f\|_\rho = 1} \|Lf\|_H$. We will use the conventional notations on summation and production: $\prod_{i=t+1}^t = 1$ and $\sum_{i=t+1}^t = 0$.

We next introduce some auxiliary operators. Let $\mathcal{S}_\rho : H \to L^2(H, \rho_X)$ be the linear map $\omega \to \langle \omega, \cdot \rangle_H$, which is bounded by $\kappa$ under Assumption (3). Furthermore, we consider the adjoint operator $\mathcal{S}_\rho^* : L^2(H, \rho_X) \to H$, the covariance operator $\mathcal{T} : H \to H$ given by $\mathcal{T} = \mathcal{S}_\rho^* \mathcal{S}_\rho$, and the operator $\mathcal{L} : L^2(H, \rho_X) \to L^2(H, \rho_X)$ given by $\mathcal{S}_\rho \mathcal{S}_\rho^*$. It can be easily proved that $\mathcal{S}_\rho^* g = \int_X x g(x) d\rho_X(x)$ and $\mathcal{T} = \int_X \langle \cdot, x \rangle_H x d\rho_X(x)$. The operators $\mathcal{T}$ and $\mathcal{L}$ can be proved to be positive trace class operators (and hence compact). For any $\omega \in H$, it is easy to prove the following isometry property [25]

$$\|\mathcal{S}_\rho \omega\|_\rho = \|\sqrt{\mathcal{T}} \omega\|_H. \tag{17}$$

We define the sampling operator $\mathcal{S}_{\mathbf{x}} : H \to \mathbb{R}^m$ by $(\mathcal{S}_{\mathbf{x}} \omega)_i = \langle \omega, x_i \rangle_H$, $i \in [m]$, where the norm $\| \cdot \|_{\mathbb{R}^m}$ in $\mathbb{R}^m$ is the Euclidean norm times $1/m$. Its adjoint operator $\mathcal{S}_{\mathbf{x}}^* : \mathbb{R}^m \to H$, defined by $\langle \mathcal{S}_{\mathbf{x}}^* \mathbf{y}, \omega \rangle_H = \langle \mathbf{y}, \mathcal{S}_{\mathbf{x}} \omega \rangle_{\mathbb{R}^m}$ for $\mathbf{y} \in \mathbb{R}^m$ is thus given by $\mathcal{S}_{\mathbf{x}}^* \mathbf{y} = \frac{1}{m} \sum_{i=1}^m y_i x_i$. Moreover, we can define the empirical covariance operator $\mathcal{T}_{\mathbf{x}} : H \to H$ such that $\mathcal{T}_{\mathbf{x}} = \mathcal{S}_{\mathbf{x}}^* \mathcal{S}_{\mathbf{x}}$. Obviously,

$$\mathcal{T}_{\mathbf{x}} = \frac{1}{m} \sum_{i=1}^m \langle \cdot, x_i \rangle_H x_i.$$

With these notations, (13) and (14) can be rewritten as

$$\mu_{t+1} = \mu_t - \eta_t (\mathcal{T} \mu_t - \mathcal{S}_\rho^* f_\rho), \qquad t = 1, \dots, T, \tag{18}$$

and

$$\nu_{t+1} = \nu_t - \eta_t (\mathcal{T}_{\mathbf{x}} \nu_t - \mathcal{S}_{\mathbf{x}}^* \mathbf{y}), \qquad t = 1, \dots, T, \tag{19}$$

respectively.

Using the projection theorem, one can prove that

$$\mathcal{S}_\rho^* f_\rho = \mathcal{S}_\rho^* f_\mathcal{H}. \tag{20}$$

Indeed, since $f_\mathcal{H}$ is the projection of the regression function $f_\rho$ onto the closer of $H_\rho$ in $L^2(H, \rho_X)$, according to the projection theorem, one has

$$\langle f_\mathcal{H} - f_\rho, \mathcal{S}_\rho \omega \rangle_\rho = 0, \qquad \forall \omega \in H,$$

which can be written as

$$\langle \mathcal{S}_\rho^* f_\mathcal{H} - \mathcal{S}_\rho^* f_\rho, \omega \rangle_H = 0, \qquad \forall \omega \in H,$$

and thus leads to (20).

### A.2  Concentration Inequality

We need the following concentration result for Hilbert space valued random variable used in Caponnetto and De Vito [4] and based on the results in Pinelis and Sakhanenko [17].

**Lemma A.1.** *Let $w_1, \cdots, w_m$ be i.i.d random variables in a Hilbert space with norm $\| \cdot \|$. Suppose that there are two positive constants $B$ and $\sigma^2$ such that*

$$\mathbb{E}[\|w_1 - \mathbb{E}[w_1]\|^l] \leq \frac{1}{2} l! B^{l-2} \sigma^2, \quad \forall l \geq 2. \tag{21}$$

*Then for any $0 < \delta < 1$, the following holds with probability at least $1 - \delta$,*

$$\left\| \frac{1}{m} \sum_{k=1}^{m} w_m - \mathbb{E}[w_1] \right\| \le 2 \left( \frac{B}{m} + \frac{\sigma}{\sqrt{m}} \right) \log \frac{2}{\delta}.$$

*In particular, (21) holds if*

$$\|w_1\| \le B/2 \ \ a.s., \quad and \ \mathbb{E}[\|w_1\|^2] \le \sigma^2. \tag{22}$$

### A.3 Basic Estimates

**Lemma A.2.** *Let $\theta \in [0, 1[$, and $t \in \mathbb{N}$. Then*

$$\frac{t^{1-\theta}}{2} \le \sum_{k=1}^{t} k^{-\theta} \le \frac{t^{1-\theta}}{1-\theta}.$$

*Proof.* Note that

$$\sum_{k=1}^{t} k^{-\theta} \le 1 + \sum_{k=2}^{t} \int_{k-1}^{k} u^{-\theta} du = 1 + \int_{1}^{t} u^{-\theta} du = \frac{t^{1-\theta} - \theta}{1-\theta},$$

which leads to the first part of the desired result. Similarly,

$$\sum_{k=1}^{t} k^{-\theta} \ge \sum_{k=1}^{t} \int_{k}^{k+1} u^{-\theta} du = \int_{1}^{t+1} u^{-\theta} du = \frac{(t+1)^{1-\theta} - 1}{1-\theta},$$

and by mean value theorem, $(t+1)^{1-\theta} - 1 \ge (1-\theta)t(t+1)^{-\theta} \ge (1-\theta)t^{1-\theta}/2$. This proves the second part of the desired result. The proof is complete. $\square$

**Lemma A.3.** *Let $\theta \in \mathbb{R}$ and $t \in \mathbb{N}$. Then*

$$\sum_{k=1}^{t} k^{-\theta} \le t^{\max(1-\theta,0)}(1 + \log t).$$

*Proof.* Note that

$$\sum_{k=1}^{t} k^{-\theta} = \sum_{k=1}^{t} k^{-1} k^{1-\theta} \le t^{\max(1-\theta,0)} \sum_{k=1}^{t} k^{-1},$$

and

$$\sum_{k=1}^{t} k^{-1} \le 1 + \sum_{k=2}^{t} \int_{k-1}^{k} u^{-1} du = 1 + \log t.$$

$\square$

**Lemma A.4.** *Let $q \in \mathbb{R}$ and $t \in \mathbb{N}$ with $t \ge 3$. Then*

$$\sum_{k=1}^{t-1} \frac{1}{t-k} k^{-q} \le 2 t^{-\min(q,1)}(1 + \log t).$$

*Proof.* Note that

$$\sum_{k=1}^{t-1} \frac{1}{t-k} k^{-q} = \sum_{k=1}^{t-1} \frac{k^{1-q}}{(t-k)k} \le t^{\max(1-q,0)} \sum_{k=1}^{t-1} \frac{1}{(t-k)k},$$

and that by Lemma A.3,

$$\sum_{k=1}^{t-1} \frac{1}{(t-k)k} = \frac{1}{t} \sum_{k=1}^{t-1} \left( \frac{1}{t-k} + \frac{1}{k} \right) = \frac{2}{t} \sum_{k=1}^{t-1} \frac{1}{k} \le \frac{2}{t}(1 + \log t).$$

$\square$

## B    Bias

In this section, we develop upper bounds for the bias, i.e., $\|\mathcal{S}_\rho \mu_t - f_\mathcal{H}\|_\rho^2$. Towards this end, we introduce the following lemma, whose proof borrows idea from [29, 26].

**Lemma B.1.** *Let $L$ be a compact positive operator on a separable Hilbert space $H$. Assume that $\eta_1 \|L\| \leq 1$. Then for $t \in \mathbb{N}$ and any non-negative integer $k \leq t - 1$,*

$$\|\Pi_{k+1}^t(L)L^\zeta\| \leq \left( \frac{\zeta}{\mathrm{e} \sum_{j=k+1}^t \eta_j} \right)^\zeta . \tag{23}$$

*Proof.* Let $\{\sigma_i\}$ be the sequence of eigenvalues of $L$. We have

$$\|\Pi_{k+1}^t(L)L^\zeta\| = \sup_i \prod_{l=k+1}^t (1 - \eta_l \sigma_i)\sigma_i^\zeta .$$

Using the basic inequality

$$1 + x \leq \mathrm{e}^x \qquad \text{for all } x \geq -1, \tag{24}$$

with $\eta_l \|L\| \leq 1$, we get

$$
\begin{aligned}
\|\Pi_{k+1}^t(L)L^\zeta\| &\leq \sup_i \exp\left\{ -\sigma_i \sum_{l=k+1}^t \eta_l \right\} \sigma_i^\zeta \\
&\leq \sup_{x \geq 0} \exp\left\{ -x \sum_{l=k+1}^t \eta_l \right\} x^\zeta .
\end{aligned}
$$

The maximum of the function $g(x) = \mathrm{e}^{-cx} x^\zeta$ ( with $c > 0$) over $\mathbb{R}_+$ is achieved at $x_{\max} = \zeta/c$, and thus

$$\sup_{x \geq 0} \mathrm{e}^{-cx} x^\zeta = \left( \frac{\zeta}{\mathrm{e}c} \right)^\zeta . \tag{25}$$

Using this inequality, one can get the desired result (23). $\qquad\square$

With the above lemma and Lemma A.2 from the appendix, we can derive the following result for the bias.

**Proposition B.2.** *Under Assumption 2, let $\eta_1 \kappa^2 \leq 1$. Then, for any $t \in \mathbb{N}$,*

$$\|\mathcal{S}_\rho \mu_{t+1} - f_\mathcal{H}\|_\rho \leq R \left( \frac{\zeta}{2 \sum_{j=1}^t \eta_j} \right)^\zeta . \tag{26}$$

*In particular, if $\eta_t = \eta t^{-\theta}$ for all $t \in \mathbb{N}$, with $\eta \in ]0, \kappa^{-2}]$ and $\theta \in [0, 1[$, then*

$$\|\mathcal{S}_\rho \mu_{t+1} - f_\mathcal{H}\|_\rho \leq R\zeta^\zeta \eta^{-\zeta} t^{(\theta-1)\zeta} . \tag{27}$$

*Proof.* The result is essentially proved in [28], see also [19]. For the sake of completeness, we provide a proof here. Since $\mu_{t+1}$ is given by (18), introducing with (20),

$$\mu_{t+1} = \mu_t - \eta_t(\mathcal{T}\mu_t - \mathcal{S}_\rho^* f_\mathcal{H}). \tag{28}$$

Thus,

$$\mathcal{S}_\rho \mu_{t+1} = \mathcal{S}_\rho \mu_t - \eta_t \mathcal{S}_\rho(\mathcal{T}\mu_t - \mathcal{S}_\rho^* f_\mathcal{H}) = \mathcal{S}_\rho \mu_t - \eta_t \mathcal{L}(\mathcal{S}_\rho \mu_t - f_\mathcal{H}). \tag{29}$$

Subtracting both sides by $f_\mathcal{H}$,

$$\mathcal{S}_\rho \mu_{t+1} - f_\mathcal{H} = (I - \eta_t \mathcal{L})(\mathcal{S}_\rho \mu_t - f_\mathcal{H}).$$

Using this equality iteratively, with $\mu_1 = 0$,

$$\mathcal{S}_\rho \mu_{t+1} - f_\mathcal{H} = -\Pi_1^t(\mathcal{L})f_\mathcal{H}.$$

Taking the $L^2(H, \rho_X)$-norm, by Assumption 2,

$$\|\mathcal{S}_\rho \mu_{t+1} - f_\mathcal{H}\|_\rho = \|\Pi_1^t(\mathcal{L})f_\mathcal{H}\|_\rho \leq \|\Pi_1^t(\mathcal{L})\mathcal{L}^\zeta\| R.$$

By applying Lemma B.1, we get (26). Combining (26) with Lemma A.2, we get (27). The proof is complete. $\qquad\square$

The following lemma gives upper bounds for the sequence $\{\mu_t\}_{t \in \mathbb{N}}$ in $H$-norm. It will be used for the estimation on the sample variance in the next section.

**Lemma B.3.** *Under Assumption 2, the following holds for all $t \in \mathbb{N}$:*
*1) If $\zeta \geq 1/2$,*

$$\|\mu_t\|_H \leq R\kappa^{2\zeta-1}. \tag{30}$$

*2) If $\zeta \in ]0, 1/2]$,*

$$\|\mu_t\|_H \leq \kappa^{2\zeta-1} \vee \left(\sum_{k=1}^t \eta_k\right)^{\frac{1}{2}-\zeta}. \tag{31}$$

*Proof.* The proof for the fixed step-size can be found in [19]. Following from (28), we have

$$\mu_{t+1} = (I - \eta_t\mathcal{T})\mu_t + \eta_t\mathcal{S}_\rho^* f_{\mathcal{H}}.$$

Applying this relationship iteratively, and introducing with $\mu_1 = 0$, we get

$$\mu_{t+1} = \sum_{k=1}^t \eta_k \Pi_{k+1}^t(\mathcal{T})\mathcal{S}_\rho^* f_{\mathcal{H}} = \sum_{k=1}^t \eta_k \mathcal{S}_\rho^* \Pi_{k+1}^t(\mathcal{L}) f_{\mathcal{H}}.$$

Therefore, using Assumption 2 and the spectrum theory,

$$\|\mu_{t+1}\|_H \leq \left\|\sum_{k=1}^t \eta_k \mathcal{S}_\rho^* \Pi_{k+1}^t(\mathcal{L})\mathcal{L}^\zeta\right\| R \leq R \max_{\sigma \in ]0,\kappa^2]} \sigma^{1/2+\zeta} \sum_{k=1}^t \eta_k \Pi_{k+1}^t(\sigma).$$

If $\zeta \geq 1/2$, for any $\sigma \in ]0, \kappa^2]$,

$$\sigma^{1/2+\zeta} \sum_{k=1}^t \eta_k \Pi_{k+1}^t(\sigma) \leq \kappa^{2\zeta-1}\sigma \sum_{k=1}^t \eta_k \Pi_{k+1}^t(\sigma) \leq \kappa^{2\zeta-1},$$

where for the last inequality, we used

$$\sum_{k=1}^t \eta_k \sigma \Pi_{k+1}^t(\sigma) = \sum_{k=1}^t (1 - (1 - \eta_k\sigma))\Pi_{k+1}^t(\sigma) = \sum_{k=1}^t \Pi_{k+1}^t(\sigma) - \sum_{k=1}^t \Pi_k^t(\sigma) = 1 - \Pi_1^t(\sigma).$$

Thus,

$$\|\mu_{t+1}\|_H \leq R\kappa^{2\zeta-1}.$$

The case for $\zeta < 1/2$ is similar to that in [19]. We omit it. The proof is complete. $\square$

## C   Sample Variance

In this section, we aim to estimate the sample variance, i.e., $\mathbb{E}[\|\mathcal{S}_\rho\mu_t - \mathcal{S}_\rho\nu_t\|_\rho^2]$. Towards this end, we need some preliminary analysis. We first introduce the following key inequality, which provides the hinge idea on estimating $\mathbb{E}[\|\mathcal{S}_\rho\mu_t - \mathcal{S}_\rho\nu_t\|_\rho^2]$.

**Lemma C.1.** *For all $t \in [T]$, we have*

$$\|\mathcal{S}_\rho\nu_{t+1} - \mathcal{S}_\rho\mu_{t+1}\|_\rho \leq \sum_{k=1}^t \eta_k \left\|\mathcal{T}^{\frac{1}{2}}\Pi_{k+1}^t(\mathcal{T}_\mathbf{x})N_k\right\|_K, \tag{32}$$

*where*

$$N_k = (\mathcal{T}\mu_k - \mathcal{S}_\rho^* f_\rho) - (\mathcal{T}_\mathbf{x}\mu_k - \mathcal{S}_\mathbf{x}^*\mathbf{y}), \qquad \forall k \in [T]. \tag{33}$$

*Proof.* Since $\nu_{t+1}$ and $\mu_{t+1}$ are given by (19) and (18), respectively,

$$\begin{aligned}
\nu_{t+1} - \mu_{t+1} &= \nu_t - \mu_t + \eta_t\left\{(\mathcal{T}\mu_t - \mathcal{S}_\rho^* f_\rho) - (\mathcal{T}_\mathbf{x}\nu_t - \mathcal{S}_\mathbf{x}^*\mathbf{y})\right\} \\
&= (I - \eta_t\mathcal{T}_\mathbf{x})(\nu_t - \mu_t) + \eta_t\left\{(\mathcal{T}\mu_t - \mathcal{S}_\rho^* f_\rho) - (\mathcal{T}_\mathbf{x}\mu_t - \mathcal{S}_\mathbf{x}^*\mathbf{y})\right\},
\end{aligned}$$

which is exactly

$$\nu_{t+1} - \mu_{t+1} = (I - \eta_t \mathcal{T}_{\mathbf{x}})(\nu_t - \mu_t) + \eta_t N_t.$$

Applying this relationship iteratively, with $\nu_1 = \mu_1 = 0$,

$$\nu_{t+1} - \mu_{t+1} = \Pi_1^t(\mathcal{T}_{\mathbf{x}})(\nu_1 - \mu_1) + \sum_{k=1}^{t} \eta_k \Pi_{k+1}^t(\mathcal{T}_{\mathbf{x}}) N_k = \sum_{k=1}^{t} \eta_k \Pi_{k+1}^t(\mathcal{T}_{\mathbf{x}}) N_k.$$

By (17), we have

$$\|\mathcal{S}_\rho \nu_{t+1} - \mathcal{S}_\rho \mu_{t+1}\|_\rho = \left\| \sum_{k=1}^{t} \eta_k \mathcal{T}^{\frac{1}{2}} \Pi_{k+1}^t(\mathcal{T}_{\mathbf{x}}) N_k \right\|_H,$$

which leads to the desired result (32). The proof is complete. $\qquad\square$

The above lemma demonstrates that in order to upper bound $\mathbb{E}[\|\mathcal{S}_\rho \mu_t - \mathcal{S}_\rho \nu_t\|_\rho^2]$, one may only need to bound $\left\| \mathcal{T}^{\frac{1}{2}} \Pi_{k+1}^t(\mathcal{T}_{\mathbf{x}}) N_k \right\|_H$. A detailed look at this latter term indicates that one may analysis the terms $\mathcal{T}^{\frac{1}{2}} \Pi_{k+1}^t(\mathcal{T}_{\mathbf{x}})$ and $N_k$ separately, since $\mathbb{E}[N_k] = 0$ and the properties of the deterministic sequence $\{\mu_k\}_k$ are well developed in Section B.

**Lemma C.2.** *Under Assumptions 2 and 3 , let $\zeta \geq 1/2$. Then for any fixed $\lambda > 0$, with probability at least $1 - \delta_1$, the following holds for all $k \in \mathbb{N}$ :*
*1) If $\zeta \geq 1/2$,*

$$\|(\mathcal{T} + \lambda)^{-\frac{1}{2}} N_k\|_H \leq 4(R\kappa^{2\zeta} + \sqrt{M}) \left( \frac{\kappa}{m\sqrt{\lambda}} + \frac{\sqrt{2\sqrt{v}c_\gamma}}{\sqrt{m\lambda^\gamma}} \right) \log \frac{4}{\delta_1}. \tag{34}$$

*2) If $\zeta \in ]0, 1/2]$,*

$$\|(\mathcal{T} + \lambda)^{-\frac{1}{2}} N_k\|_H \leq 4 \left( \kappa \left( \kappa^{2\zeta-1} \vee \left( \sum_{i=1}^{k} \eta_i \right)^{\frac{1}{2}-\zeta} \right) + \sqrt{M} \right) \left( \frac{\kappa}{m\sqrt{\lambda}} + \frac{\sqrt{2\sqrt{v}c_\gamma}}{\sqrt{m\lambda^\gamma}} \right) \log \frac{4}{\delta_1}. \tag{35}$$

*Proof.* We will apply Berstein inequality from Lemma A.1 to prove the result.

*Bounding* $\left\| (\mathcal{T} + \lambda)^{-\frac{1}{2}} \left( \mathcal{S}_\rho^* f_\rho - \mathcal{S}_{\mathbf{x}}^* \mathbf{y} \right) \right\|_H$

For all $i \in [m]$, let $w_i = y_i (\mathcal{T} + \lambda I)^{-\frac{1}{2}} x_i$. Obviously, from the definitions of $f_\rho$ (see (6)) and $\mathcal{S}_\rho$,

$$\mathbb{E}[w_1] = \mathbb{E}_{x_1}[f_\rho(x_1)(\mathcal{T} + \lambda I)^{-\frac{1}{2}} x_1] = (\mathcal{T} + \lambda I)^{-\frac{1}{2}} \mathcal{S}_\rho^* f_\rho.$$

Thus,

$$(\mathcal{T} + \lambda)^{-\frac{1}{2}} \left( \mathcal{S}_\rho^* f_\rho - \mathcal{S}_{\mathbf{x}}^* \mathbf{y} \right) = \frac{1}{m} \sum_{i=1}^{\infty} (\mathbb{E}[w_i] - w_i).$$

We next estimate the constants $B$ and $\sigma^2(w_1)$ in (21). Note that for any $l \geq 2$,

$$\mathbb{E}[\|w_1 - \mathbb{E}[w_1]\|_H^l] \leq \mathbb{E}[(\|w_1\|_H + \mathbb{E}[\|w_1\|_H])^l].$$

By using Hölder's inequality twice,

$$\mathbb{E}[\|w_1 - \mathbb{E}[w_1]\|_H^l] \leq 2^{l-1} \mathbb{E}[\|w_1\|_H^l + (\mathbb{E}[\|w_1\|_H])^l] \leq 2^{l-1} \mathbb{E}[\|w_1\|_H^l + \mathbb{E}[\|w_1\|_H^l]].$$

The right-hand side is exactly $2^l \mathbb{E}[\|w_1\|_H^l]$. Therefore, by recalling the definition of $w_1$ and expanding the integration,

$$\mathbb{E}[\|w_1 - \mathbb{E}[w_1]\|_H^l] \leq 2^l \int_Y y^l d\rho(y|x) \int_X \|(\mathcal{T} + \lambda I)^{-\frac{1}{2}} x\|_H^l d\rho_X(x). \tag{36}$$

Note that by using Hölder's inequality,

$$\int_Y y^l d\rho(y|x) \int_X \leq \left( \int_Y |y|^{2l} d\rho(y|x) \right)^{\frac{1}{2}}.$$

Using Assumption 1 to the above,

$$\int_Y y^l d\rho(y|x) \int_X \leq \sqrt{l! M^l v} \leq l! (\sqrt{M})^l \sqrt{v}.$$

Plugging the above into (36), we reach

$$\mathbb{E}[\|w_1 - \mathbb{E}[w_1]\|_H^l] \leq l! (2\sqrt{M})^l \sqrt{v} \int_X \|(\mathcal{T} + \lambda I)^{-\frac{1}{2}} x\|_H^l d\rho_X(x).$$

Using Assumption (3) which imples

$$\|(\mathcal{T} + \lambda I)^{-\frac{1}{2}} x\|_H \leq \frac{\|x\|_H}{\sqrt{\lambda}} \leq \frac{\kappa}{\sqrt{\lambda}},$$

we get that

$$\mathbb{E}[\|w_1 - \mathbb{E}[w_1]\|_K^l] \leq l! (2\sqrt{M})^l \sqrt{v} \left( \frac{\kappa}{\sqrt{\lambda}} \right)^{l-2} \int_X \|(\mathcal{T} + \lambda I)^{-\frac{1}{2}} x\|_H^2 d\rho_X(x).$$

Using the fact that $\mathbb{E}[\|\xi\|_H^2] = \mathbb{E}[\text{tr}(\xi \otimes \xi)] = \text{tr}(\mathbb{E}[\xi \otimes \xi])$ and $\mathbb{E}[x \otimes x] = \mathcal{T}$, we know that

$$\int_X \|(\mathcal{T} + \lambda I)^{-\frac{1}{2}} x\|_H^2 d\rho_X(x) = \text{tr}((\mathcal{T} + \lambda I)^{-\frac{1}{2}} \mathcal{T}(\mathcal{T} + \lambda I)^{-\frac{1}{2}}) = \text{tr}((\mathcal{T} + \lambda I)^{-1} \mathcal{T}),$$

and as a result of the above and Assumption 3,

$$\int_X \|(\mathcal{T} + \lambda I)^{-\frac{1}{2}} x\|_H^2 d\rho_X(x) \leq c_\gamma \lambda^{-\gamma}.$$

Therefore,

$$\mathbb{E}[\|w_1 - \mathbb{E}[w_1]\|_H^l] \leq l! (2\sqrt{M})^l \sqrt{v} \left( \frac{\kappa}{\sqrt{\lambda}} \right)^{l-2} c_\gamma \lambda^{-\gamma} = \frac{1}{2} l! \left( \frac{2\kappa\sqrt{M}}{\sqrt{\lambda}} \right)^{l-2} 8M\sqrt{v} c_\gamma \lambda^{-\gamma}.$$

Applying Berstein inequality with $B = \frac{2\kappa\sqrt{M}}{\sqrt{\lambda}}$ and $\sigma = \sqrt{8M\sqrt{v} c_\gamma \lambda^{-\gamma}}$, we get that with probability at least $1 - \frac{\delta_1}{2}$, there holds

$$\left\| (\mathcal{T} + \lambda)^{-\frac{1}{2}} \left( \mathcal{S}_\rho^* f_\rho - \mathcal{S}_{\mathbf{x}}^* \mathbf{y} \right) \right\|_H = \left\| \frac{1}{m} \sum_{i=1}^m (\mathbb{E}[w_i] - w_i) \right\|_H \leq 4\sqrt{M} \left( \frac{\kappa}{m\sqrt{\lambda}} + \frac{\sqrt{2\sqrt{v} c_\gamma}}{\sqrt{m\lambda^\gamma}} \right) \log \frac{4}{\delta_1}. \tag{37}$$

*Bounding* $\|(\mathcal{T} + \lambda)^{-\frac{1}{2}}(\mathcal{T} - \mathcal{T}_{\mathbf{x}})\|$
Let $\xi_i = (\mathcal{T} + \lambda)^{-\frac{1}{2}} x_i \otimes x_i$, for all $i \in [m]$. It is easy to see that $\mathbb{E}[\xi_i] = (\mathcal{T} + \lambda)^{-\frac{1}{2}} \mathcal{T}$, and that $(\mathcal{T} + \lambda)^{-\frac{1}{2}}(\mathcal{T} - \mathcal{T}_{\mathbf{x}}) = \frac{1}{m} \sum_{i=1}^m (\mathbb{E}[\xi_i] - \xi_i)$. Denote the Hilbert-Schmidt norm of a bounded operator from $H$ to $H$ by $\| \cdot \|_{HS}$. Note that

$$\|\xi_1\|_{HS}^2 = \|x_1\|_H^2 \text{Trace}((\mathcal{T} + \lambda)^{-1/2} x_1 \otimes x_1 (\mathcal{T} + \lambda)^{-1/2}) = \|x_1\|_H^2 \text{Trace}((\mathcal{T} + \lambda)^{-1} x_1 \otimes x_1).$$

By Assumption (3),

$$\|\xi_1\|_{HS} \leq \sqrt{\kappa^2 \text{Trace}((\mathcal{T} + \lambda)^{-1} x_1 \otimes x_1)} \leq \sqrt{\kappa^2 \text{Trace}(x_1 \otimes x_1)/\lambda} \leq \kappa^2/\sqrt{\lambda},$$

and furthermore, by Assumption 3,

$$\mathbb{E}[\|\xi_1\|_{HS}^2] \leq \kappa^2 \mathbb{E}\text{Trace}((\mathcal{T} + \lambda)^{-1} x_1 \otimes x_1) = \kappa^2 \text{Trace}((\mathcal{T} + \lambda)^{-1} \mathcal{T}) \leq \kappa^2 c_\gamma \lambda^{-\gamma}.$$

According to Lemma A.1, we get that with probability at least $1 - \frac{\delta_1}{2}$, there holds

$$\|(\mathcal{T} + \lambda)^{-\frac{1}{2}}(\mathcal{T} - \mathcal{T}_{\mathbf{x}})\|_{HS} \leq 2\kappa \left( \frac{2\kappa}{m\sqrt{\lambda}} + \frac{\sqrt{c_\gamma}}{\sqrt{m\lambda^\gamma}} \right) \log \frac{4}{\delta_1}. \tag{38}$$

Finally, using the triangle inequality, we have,

$$\|(\mathcal{T} + \lambda)^{-\frac{1}{2}} N_k\|_H \leq \|(\mathcal{T} + \lambda)^{-\frac{1}{2}}(\mathcal{T} - \mathcal{T}_{\mathbf{x}})\| \|\mu_k\|_H + \left\| (\mathcal{T} + \lambda)^{-\frac{1}{2}} \left( \mathcal{S}_\rho^* f_\rho - \mathcal{S}_{\mathbf{x}}^* \mathbf{y} \right) \right\|_H.$$

Applying Lemma B.3 to the above, introducing with (37) and (38), and then noting that $\kappa \geq 1$ and $v \geq 1$, one can prove the desired results. $\quad\square$

The next lemma is borrowed from [20], derived by applying a recent Bernstein inequality from [27, 13] for a sum of random operators.

**Lemma C.3.** *Let $\delta_2 \in (0,1)$ and $\frac{9\kappa^2}{m} \log \frac{m}{\delta_2} \leq \lambda \leq \|\mathcal{T}\|$. Then the following holds with probability at least $1 - \delta_2$,*

$$\|(\mathcal{T}_\mathbf{x} + \lambda I)^{-\frac{1}{2}} \mathcal{T}^{\frac{1}{2}}\| \leq \|(\mathcal{T}_\mathbf{x} + \lambda)^{-\frac{1}{2}}(\mathcal{T} + \lambda)^{\frac{1}{2}}\| \leq 2. \qquad (39)$$

Now we are in a position to estimate the sample variance.

**Proposition C.4.** *Let $\eta_1 \kappa^2 \leq 1$ and (34) for all $k \in [T]$. Assume that (39) holds. Then the following holds for all $t \in [T]$ :*
*1) If $\zeta \geq 1/2$,*

$$\|\mathcal{S}_\rho \nu_{t+1} - \mathcal{S}_\rho \mu_{t+1}\|_\rho$$

$$\leq 4(R\kappa^{2\zeta} + \sqrt{M}) \left( \frac{\kappa}{m\sqrt{\lambda}} + \frac{\sqrt{2\sqrt{v}c_\gamma}}{\sqrt{m\lambda^\gamma}} \right) \left( \sum_{k=1}^{t-1} \frac{\eta_k/2}{\sum_{i=k+1}^{t} \eta_i} + \lambda \sum_{k=1}^{t-1} \eta_k + \sqrt{2}\kappa^2 \eta_t \right) \log \frac{4}{\delta_1}. \qquad (40)$$

*2) If $\zeta \leq 1/2$,*

$$\|\mathcal{S}_\rho \nu_{t+1} - \mathcal{S}_\rho \mu_{t+1}\|_\rho \leq 4 \left( \kappa \left( \kappa^{2\zeta-1} \vee \left( \sum_{i=1}^{k} \eta_i \right)^{\frac{1}{2}-\zeta} \right) + \sqrt{M} \right)$$

$$\times \left( \sum_{k=1}^{t-1} \frac{\eta_k/2}{\sum_{i=k+1}^{t} \eta_i} + \lambda \sum_{k=1}^{t-1} \eta_k + \sqrt{2}\kappa^2 \eta_t \right) \left( \frac{\kappa}{m\sqrt{\lambda}} + \frac{\sqrt{2\sqrt{v}c_\gamma}}{\sqrt{m\lambda^\gamma}} \right) \log \frac{4}{\delta_1}. \qquad (41)$$

*Proof.* For notational simplicity, we let $\mathcal{T}_\lambda = \mathcal{T} + \lambda I$ and $\mathcal{T}_{\mathbf{x},\lambda} = \mathcal{T}_\mathbf{x} + \lambda I$. Note that by Lemma C.1, we have (32). When $k \in [t-1]$, by rewriting $\mathcal{T}^{\frac{1}{2}} \Pi_{k+1}^{t}(\mathcal{T}_\mathbf{x}) N_k$ as

$$\mathcal{T}^{\frac{1}{2}} \mathcal{T}_{\mathbf{x},\lambda}^{-\frac{1}{2}} \mathcal{T}_{\mathbf{x},\lambda}^{\frac{1}{2}} \Pi_{k+1}^{t}(\mathcal{T}_\mathbf{x}) \mathcal{T}_{\mathbf{x},\lambda}^{\frac{1}{2}} \mathcal{T}_{\mathbf{x},\lambda}^{-\frac{1}{2}} \mathcal{T}_\lambda^{\frac{1}{2}} \mathcal{T}_\lambda^{-\frac{1}{2}} N_k,$$

we can upper bound $\|\mathcal{T}^{\frac{1}{2}} \Pi_{k+1}^{t}(\mathcal{T}_\mathbf{x}) N_k\|_H$ as

$$\|\mathcal{T}^{\frac{1}{2}} \Pi_{k+1}^{t}(\mathcal{T}_\mathbf{x}) N_k\|_H \leq \|\mathcal{T}^{\frac{1}{2}} \mathcal{T}_{\mathbf{x},\lambda}^{-\frac{1}{2}}\| \|\mathcal{T}_{\mathbf{x},\lambda}^{\frac{1}{2}} \Pi_{k+1}^{t}(\mathcal{T}_\mathbf{x}) \mathcal{T}_{\mathbf{x},\lambda}^{\frac{1}{2}}\| \|\mathcal{T}_{\mathbf{x},\lambda}^{-\frac{1}{2}} \mathcal{T}_\lambda^{\frac{1}{2}}\| \|\mathcal{T}_\lambda^{-\frac{1}{2}} N_k\|_H.$$

Applying (39), the above can be relaxed as

$$\|\mathcal{T}^{\frac{1}{2}} \Pi_{k+1}^{t}(\mathcal{T}_\mathbf{x}) N_k\|_H \leq 4\|\mathcal{T}_{\mathbf{x},\lambda}^{\frac{1}{2}} \Pi_{k+1}^{t}(\mathcal{T}_\mathbf{x}) \mathcal{T}_{\mathbf{x},\lambda}^{\frac{1}{2}}\| \|\mathcal{T}_\lambda^{-\frac{1}{2}} N_k\|_H,$$

which is equivalent to

$$\|\mathcal{T}_\lambda^{\frac{1}{2}} \Pi_{k+1}^{t}(\mathcal{T}_\mathbf{x}) N_k\|_H \leq 4\|\mathcal{T}_{\mathbf{x},\lambda} \Pi_{k+1}^{t}(\mathcal{T}_\mathbf{x})\| \|\mathcal{T}_\lambda^{-\frac{1}{2}} N_k\|_H.$$

Thus, following from $\eta_k \kappa^2 \leq 1$ which implies $\eta_k \|\mathcal{T}_\mathbf{x}\| \leq 1$,

$$\|\mathcal{T}_{\mathbf{x},\lambda} \Pi_{k+1}^{t}(\mathcal{T}_\mathbf{x})\| \leq \|\mathcal{T}_\mathbf{x} \Pi_{k+1}^{t}(\mathcal{T}_\mathbf{x})\| + \|\lambda \Pi_{k+1}^{t}(\mathcal{T}_\mathbf{x})\|$$
$$\leq \|\mathcal{T}_\mathbf{x} \Pi_{k+1}^{t}(\mathcal{T}_\mathbf{x})\| + \lambda.$$

Applying Lemma B.1 with $\zeta = 1$ to bound $\|\mathcal{T}_\mathbf{x} \Pi_{k+1}^{t}(\mathcal{T}_\mathbf{x})\|$, we get

$$\|\mathcal{T}_{\mathbf{x},\lambda} \Pi_{k+1}^{t}(\mathcal{T}_\mathbf{x})\| \leq \frac{1}{e \sum_{j=k+1}^{t} \eta_j} + \lambda.$$

When $k = t$,

$$\|\mathcal{T}^{\frac{1}{2}} \Pi_{k+1}^{t}(\mathcal{T}_\mathbf{x}) N_k\|_H = \|\mathcal{T}^{\frac{1}{2}} N_t\|_H \leq \|\mathcal{T}^{\frac{1}{2}}\| \|\mathcal{T}_\lambda^{\frac{1}{2}}\| \|\mathcal{T}_\lambda^{-\frac{1}{2}} N_t\|_H$$
$$\leq \|\mathcal{T}\|^{\frac{1}{2}} (\|\mathcal{T}\| + \lambda)^{\frac{1}{2}} \|\mathcal{T}_\lambda^{-\frac{1}{2}} N_t\|_H.$$

Since $\lambda \leq \|\mathcal{T}\| \leq \text{tr}(\mathcal{T}) \leq \kappa^2$, we derive

$$\|\mathcal{T}^{\frac{1}{2}} \Pi_{k+1}^{t}(\mathcal{T}_\mathbf{x}) N_t\|_H \leq \sqrt{2}\kappa^2 \|\mathcal{T}_\lambda^{-\frac{1}{2}} N_t\|_H.$$

From the above analysis, we conclude that $\sum_{k=1}^{t} \eta_k \left\| \mathcal{T}^{\frac{1}{2}} \Pi_{k+1}^t (\mathcal{T}_{\mathbf{x}}) N_k \right\|_H$ can be upper bounded by

$$\leq \sup_{k\in[t]} \|\mathcal{T}_\lambda^{-\frac{1}{2}} N_k\|_H \left( \sum_{k=1}^{t-1} \frac{\eta_k/2}{\sum_{i=k+1}^t \eta_i} + \lambda \sum_{k=1}^{t-1} \eta_k + \sqrt{2}\kappa^2 \eta_t \right).$$

Plugging (34) (or (35)) into the above, and then combining with (32), we get the desired bound (40) (or (41)). The proof is complete. □

Setting $\eta_t = \eta_1 t^{-\theta}$ in the above proposition, with some basic estimates from Appendix A, we get the following explicit bounds for the sample variance.

**Proposition C.5.** *Let $\eta_t = \eta_1 t^{-\theta}$ and (34) for all $t \in [T]$, with $\eta_1 \in ]0, \kappa^{-2}]$ and $\theta \in [0, 1[$. Assume that (39) holds. Then the following holds for all $t \in [T]$:*
*1) If $\zeta \geq 1/2$,*

$$\|\mathcal{S}_\rho \nu_{t+1} - \mathcal{S}_\rho \mu_{t+1}\|_\rho$$
$$\leq 4(R\kappa^{2\zeta} + \sqrt{M}) \left( \frac{2\lambda \eta_1 t^{1-\theta}}{1-\theta} + \log t + 1 + \sqrt{2}\eta_1 \kappa^2 \right) \left( \frac{\kappa}{m\sqrt{\lambda}} + \frac{\sqrt{2\sqrt{v}c_\gamma}}{\sqrt{m\lambda^\gamma}} \right) \log \frac{4}{\delta_1}. \quad (42)$$

*2) If $\zeta \leq 1/2$,*

$$\|\mathcal{S}_\rho \nu_{t+1} - \mathcal{S}_\rho \mu_{t+1}\|_\rho \leq 4 \left( \kappa \left( \kappa^{2\zeta-1} \vee \left( \frac{2\eta_1 t^{1-\theta}}{1-\theta} \right)^{\frac{1}{2}-\zeta} \right) + \sqrt{M} \right)$$
$$\times \left( \frac{2\lambda \eta_1 t^{1-\theta}}{1-\theta} + \log t + 1 + \sqrt{2}\eta_1 \kappa^2 \right) \left( \frac{\kappa}{m\sqrt{\lambda}} + \frac{\sqrt{2\sqrt{v}c_\gamma}}{\sqrt{m\lambda^\gamma}} \right) \log \frac{4}{\delta_1}. \quad (43)$$

*Proof.* By Proposition C.4, we have (40). Note that

$$\sum_{k=1}^{t-1} \frac{\eta_k}{\sum_{i=k+1}^t \eta_i} = \sum_{k=1}^{t-1} \frac{k^{-\theta}}{\sum_{i=k+1}^t i^{-\theta}} \leq \sum_{k=1}^{t-1} \frac{k^{-\theta}}{(t-k)t^{-\theta}}.$$

Applying Lemma A.4, we get

$$\sum_{k=1}^{t-1} \frac{\eta_k}{\sum_{i=k+1}^t \eta_i} \leq 2 + 2\log t,$$

and by Lemma A.2,

$$\sum_{k=1}^{t-1} \eta_k = \eta_1 \sum_{k=1}^{t-1} k^{-\theta} \leq \frac{2\eta_1 t^{1-\theta}}{1-\theta}.$$

Introducing the last two estimates into (40) and (42), one can get the desired results. The proof is complete. □

In conclusion, we get the following result for the sample variance.

**Theorem C.6.** *Under Assumptions 1, 2 and 3, let $\delta_1, \delta_2 \in ]0,1[$ and $\frac{9\kappa^2}{m} \log \frac{m}{\delta_2} \leq \lambda \leq \|\mathcal{T}\|$. Let $\eta_t = \eta_1 t^{-\theta}$ for all $t \in [T]$, with $\eta_1 \in ]0, \kappa^{-2}]$ and $\theta \in [0, 1[$. Then with probability at least $1 - \delta_1 - \delta_2$, the following holds for all $t \in [T]$:*
*1) if $\zeta \geq 1/2$, we have (42).*
*2) if $\zeta < 1/2$, we have (43).*

## D  Computational Variance

In this section, we estimate the computational variance, $\mathbb{E}[\|\mathcal{S}_\rho \omega_t - \mathcal{S}_\rho \nu_t\|_\rho^2]$. For this, a series of lemmas is necessarily introduced.

## D.1 Bounding the Empirical Risk

This subsection is devoted to upper bounding $\mathbb{E}_{\mathbf{J}}[\mathcal{E}_{\mathbf{z}}(\omega_l)]$. The process relies on some tools from convex analysis and a decomposition related to the weighted averages and the last iterates from [22, 12]. We begin by introducing the following lemma, a fact based on the square loss' special properties.

**Lemma D.1.** *Given any sample* $\mathbf{z}$, *and* $l \in \mathbb{N}$, *let* $\omega \in H$ *be independent from* $\mathbf{J}_l$, *then*

$$\eta_l \left( \mathcal{E}_{\mathbf{z}}(\omega_l) - \mathcal{E}_{\mathbf{z}}(\omega) \right) \leq \|\omega_l - \omega\|_H^2 - \mathbb{E}_{\mathbf{J}_l} \|\omega_{l+1} - \omega\|_H^2 + \eta_l^2 \kappa^2 \mathcal{E}_{\mathbf{z}}(\omega_l). \tag{44}$$

*Proof.* Since $\omega_{t+1}$ is given be (4), subtracting both sides of (4) by $\omega$, taking the square $H$-norm, and expanding the inner product,

$$\|\omega_{l+1} - \omega\|_H^2 = \|\omega_l - \omega\|_H^2 + \frac{\eta_l^2}{b^2} \left\| \sum_{i=b(l-1)+1}^{bl} (\langle \omega_l, x_{j_i} \rangle_H - y_{j_i}) x_{j_i} \right\|_H^2$$

$$+ \frac{2\eta_l}{b} \sum_{i=b(l-1)+1}^{bl} (\langle \omega_l, x_{j_i} \rangle_H - y_{j_i}) \langle \omega - \omega_l, x_{j_i} \rangle_H.$$

By Assumption (3), $\|x_{j_i}\|_H \leq \kappa$, and thus

$$\left\| \sum_{i=b(l-1)+1}^{bl} (\langle \omega_l, x_{j_i} \rangle_H - y_{j_i}) x_{j_i} \right\|_H^2 \leq \left( \sum_{i=b(l-1)+1}^{bl} |\langle \omega_l, x_{j_i} \rangle_H - y_{j_i}| \kappa \right)^2$$

$$\leq \kappa^2 b \sum_{i=b(l-1)+1}^{bl} (\langle \omega_l, x_{j_i} \rangle_H - y_{j_i})^2,$$

where for the last inequality, we used Cauchy-Schwarz inequality. Thus,

$$\|\omega_{l+1} - \omega\|_H^2 \leq \|\omega_l - \omega\|_H^2 + \frac{\eta_l^2 \kappa^2}{b} \sum_{i=b(l-1)+1}^{bl} (\langle \omega_l, x_{j_i} \rangle_H - y_{j_i})^2$$

$$+ \frac{2\eta_l}{b} \sum_{i=b(l-1)+1}^{bl} (\langle \omega_l, x_{j_i} \rangle_H - y_{j_i})(\langle \omega, x_{j_i} \rangle_H - \langle \omega_l, x_{j_i} \rangle_H).$$

Using the basic inequality $a(b-a) \leq (b^2 - a^2)/2, \forall a, b \in \mathbb{R}$,

$$\|\omega_{l+1} - \omega\|_H^2 \leq \|\omega_l - \omega\|_H^2 + \frac{\eta_l^2 \kappa^2}{b} \sum_{i=b(l-1)+1}^{bl} (\langle \omega_l, x_{j_i} \rangle_H - y_{j_i})^2$$

$$+ \frac{\eta_l}{b} \sum_{i=b(l-1)+1}^{bl} \left( (\langle \omega, x_{j_i} \rangle_H - y_{j_i})^2 - (\langle \omega_l, x_{j_i} \rangle_H - y_{j_i})^2 \right).$$

Noting that $\omega_l$ and $\omega$ are independent from $\mathbf{J}_l$, and taking the expectation on both sides with respect to $\mathbf{J}_l$,

$$\mathbb{E}_{\mathbf{J}_l} \|\omega_{l+1} - \omega\|_H^2 \leq \|\omega_l - \omega\|_H^2 + \eta_l^2 \kappa^2 \mathcal{E}_{\mathbf{z}}(\omega_l) + \eta_l \left( \mathcal{E}_{\mathbf{z}}(\omega) - \mathcal{E}_{\mathbf{z}}(\omega_l) \right),$$

which leads to the desired result by rearranging terms. The proof is complete. $\square$

Using the above lemma and a decomposition related to the weighted averages and the last iterates from [22, 12], we can prove the following relationship.

**Lemma D.2.** *Let* $\eta_1 \kappa^2 \leq 1/2$ *for all* $t \in \mathbb{N}$. *Then*

$$\eta_t \mathbb{E}_{\mathbf{J}}[\mathcal{E}_{\mathbf{z}}(\omega_t)] \leq 4 \mathcal{E}_{\mathbf{z}}(0) \frac{1}{t} \sum_{l=1}^{t} \eta_l + 2\kappa^2 \sum_{k=1}^{t-1} \frac{1}{k(k+1)} \sum_{i=t-k}^{t-1} \eta_i^2 \mathbb{E}_{\mathbf{J}}[\mathcal{E}_{\mathbf{z}}(\omega_i)]. \tag{45}$$

*Proof.* For $k = 1, \cdots, t - 1$,

$$\frac{1}{k} \sum_{i=t-k+1}^{t} \eta_i \mathbb{E}_\mathbf{J}[\mathcal{E}_\mathbf{z}(\omega_i)] - \frac{1}{k+1} \sum_{i=t-k}^{t} \eta_i \mathbb{E}_\mathbf{J}[\mathcal{E}_\mathbf{z}(\omega_i)]$$

$$= \frac{1}{k(k+1)} \left\{ (k+1) \sum_{i=t-k+1}^{t} \eta_i \mathbb{E}_\mathbf{J}[\mathcal{E}_\mathbf{z}(\omega_i)] - k \sum_{i=t-k}^{t} \eta_i \mathbb{E}_\mathbf{J}[\mathcal{E}_\mathbf{z}(\omega_i)] \right\}$$

$$= \frac{1}{k(k+1)} \sum_{i=t-k+1}^{t} (\eta_i \mathbb{E}_\mathbf{J}[\mathcal{E}_\mathbf{z}(\omega_i)] - \eta_{t-k} \mathbb{E}_\mathbf{J}[\mathcal{E}_\mathbf{z}(\omega_{t-k})]).$$

Summing over $k = 1, \cdots, t - 1$, and rearranging terms, we get [12]

$$\eta_t \mathbb{E}_\mathbf{J}[\mathcal{E}_\mathbf{z}(\omega_t)] = \frac{1}{t} \sum_{i=1}^{t} \eta_i \mathbb{E}_\mathbf{J}[\mathcal{E}_\mathbf{z}(\omega_i)] + \sum_{k=1}^{t-1} \frac{1}{k(k+1)} \sum_{i=t-k+1}^{t} (\eta_i \mathbb{E}_\mathbf{J}[\mathcal{E}_\mathbf{z}(\omega_i)] - \eta_{t-k} \mathbb{E}_\mathbf{J}[\mathcal{E}_\mathbf{z}(\omega_{t-k})]).$$

Since $\{\eta_t\}_t$ is decreasing and $\mathbb{E}_\mathbf{J}[\mathcal{E}_\mathbf{z}(\omega_{t-k})]$ is non-negative, the above can be relaxed as

$$\eta_t \mathbb{E}_\mathbf{J}[\mathcal{E}_\mathbf{z}(\omega_t)] \leq \frac{1}{t} \sum_{i=1}^{t} \eta_i \mathbb{E}_\mathbf{J}[\mathcal{E}_\mathbf{z}(\omega_i)] + \sum_{k=1}^{t-1} \frac{1}{k(k+1)} \sum_{i=t-k+1}^{t} \eta_i \mathbb{E}_\mathbf{J}[\mathcal{E}_\mathbf{z}(\omega_i) - \mathcal{E}_\mathbf{z}(\omega_{t-k})]. \quad (46)$$

In the rest of the proof, we will upper bound the last two terms of the above.

To bound the first term of the right side of (46), we apply Lemma D.1 with $\omega = 0$ to get

$$\eta_l \mathbb{E}_\mathbf{J} (\mathcal{E}_\mathbf{z}(\omega_l) - \mathcal{E}_\mathbf{z}(0)) \leq \mathbb{E}_\mathbf{J}[\|\omega_l\|_H^2 - \|\omega_{l+1}\|_H^2] + \eta_l^2 \kappa^2 \mathbb{E}_\mathbf{J}[\mathcal{E}_\mathbf{z}(\omega_l)].$$

Rearranging terms,

$$\eta_l (1 - \eta_l \kappa^2) \mathbb{E}_\mathbf{J}[\mathcal{E}_\mathbf{z}(\omega_l)] \leq \mathbb{E}_\mathbf{J}[\|\omega_l\|_H^2 - \|\omega_{l+1}\|_H^2] + \eta_l \mathcal{E}_\mathbf{z}(0).$$

It thus follows from the above and $\eta_l \kappa^2 \leq 1/2$ that

$$\eta_l \mathbb{E}_\mathbf{J}[\mathcal{E}_\mathbf{z}(\omega_l)]/2 \leq \mathbb{E}_\mathbf{J}[\|\omega_l\|_H^2 - \|\omega_{l+1}\|_H^2] + \eta_l \mathcal{E}_\mathbf{z}(0).$$

Summing up over $l = 1, \cdots, t$,

$$\sum_{l=1}^{t} \eta_l \mathbb{E}_\mathbf{J}[\mathcal{E}_\mathbf{z}(\omega_l)]/2 \leq \mathbb{E}_\mathbf{J}[\|w_1\|_H^2 - \|\omega_{t+1}\|_H^2] + \mathcal{E}_\mathbf{z}(0) \sum_{l=1}^{t} \eta_l.$$

Introducing with $\omega_1 = 0$, $\|\omega_{t+1}\|_H^2 \geq 0$, and then multiplying both sides by $2/t$, we get

$$\frac{1}{t} \sum_{l=1}^{t} \eta_l \mathbb{E}_\mathbf{J}[\mathcal{E}_\mathbf{z}(\omega_l)] \leq 2 \mathcal{E}_\mathbf{z}(0) \frac{1}{t} \sum_{l=1}^{t} \eta_l. \quad (47)$$

It remains to bound the last term of (46). Let $k \in [t-1]$ and $i \in \{t-k, \cdots, t\}$. Note that given the sample $\mathbf{z}$, $\omega_i$ is depending only on $\mathbf{J}_1, \cdots, \mathbf{J}_{i-1}$ when $i > 1$ and $\omega_1 = 0$. Thus, we can apply Lemma D.1 with $\omega = \omega_{t-k}$ to derive

$$\eta_i (\mathcal{E}_\mathbf{z}(\omega_i) - \mathcal{E}_\mathbf{z}(\omega_{t-k})) \leq \|\omega_i - \omega_{t-k}\|_H^2 - \mathbb{E}_{\mathbf{J}_i} \|\omega_{i+1} - \omega_{t-k}\|_H^2 + \eta_i^2 \kappa^2 \mathcal{E}_\mathbf{z}(\omega_i).$$

Therefore,

$$\eta_i \mathbb{E}_\mathbf{J} [\mathcal{E}_\mathbf{z}(\omega_i) - \mathcal{E}_\mathbf{z}(\omega_{t-k})] \leq \mathbb{E}_\mathbf{J}[\|\omega_i - \omega_{t-k}\|_H^2 - \|\omega_{i+1} - \omega_{t-k}\|_H^2] + \eta_i^2 \kappa^2 \mathbb{E}_\mathbf{J}[\mathcal{E}_\mathbf{z}(\omega_i)].$$

Summing up over $i = t - k, \cdots, t$,

$$\sum_{i=t-k}^{t} \eta_i \mathbb{E}_\mathbf{J} [\mathcal{E}_\mathbf{z}(\omega_i) - \mathcal{E}_\mathbf{z}(\omega_{t-k})] \leq \kappa^2 \sum_{i=t-k}^{t} \eta_i^2 \mathbb{E}_\mathbf{J}[\mathcal{E}_\mathbf{z}(\omega_i)].$$

Note that the left hand side is exactly $\sum_{i=t-k+1}^{t} \eta_i \mathbb{E}_{\mathbf{J}}\left[\mathcal{E}_{\mathbf{z}}(\omega_i) - \mathcal{E}_{\mathbf{z}}(\omega_{t-k})\right]$. We thus know that the last term of (46) can be upper bounded by

$$\kappa^2 \sum_{k=1}^{t-1} \frac{1}{k(k+1)} \sum_{i=t-k}^{t} \eta_i^2 \mathbb{E}_{\mathbf{J}}[\mathcal{E}_{\mathbf{z}}(\omega_i)]$$

$$= \kappa^2 \sum_{k=1}^{t-1} \frac{1}{k(k+1)} \sum_{i=t-k}^{t-1} \eta_i^2 \mathbb{E}_{\mathbf{J}}[\mathcal{E}_{\mathbf{z}}(\omega_i)] + \kappa^2 \eta_t^2 \mathbb{E}_{\mathbf{J}}[\mathcal{E}_{\mathbf{z}}(\omega_t)] \sum_{k=1}^{t-1} \frac{1}{k(k+1)}.$$

Using the fact that

$$\sum_{k=1}^{t-1} \frac{1}{k(k+1)} = \sum_{k=1}^{t-1} \left(\frac{1}{k} - \frac{1}{k+1}\right) = 1 - \frac{1}{t} \le 1,$$

and $\kappa^2 \eta_t \le 1/2$, we get that the last term of (46) can be bounded as

$$\sum_{k=1}^{t-1} \frac{1}{k(k+1)} \sum_{i=t-k+1}^{t} \eta_i \left(\mathbb{E}_{\mathbf{J}}[\mathcal{E}_{\mathbf{z}}(\omega_i)] - \mathbb{E}_{\mathbf{J}}[\mathcal{E}_{\mathbf{z}}(\omega_{t-k})]\right)$$

$$\le \kappa^2 \sum_{k=1}^{t-1} \frac{1}{k(k+1)} \sum_{i=t-k}^{t-1} \eta_i^2 \mathbb{E}_{\mathbf{J}}[\mathcal{E}_{\mathbf{z}}(\omega_i)] + \eta_t \mathbb{E}_{\mathbf{J}}[\mathcal{E}_{\mathbf{z}}(\omega_t)]/2.$$

Plugging the above and (47) into the decomposition (46), and rearranging terms

$$\eta_t \mathbb{E}_{\mathbf{J}}[\mathcal{E}_{\mathbf{z}}(\omega_t)]/2 \le 2M^2 \frac{1}{t} \sum_{l=1}^{t} \eta_l + \kappa^2 \sum_{k=1}^{t-1} \frac{1}{k(k+1)} \sum_{i=t-k}^{t-1} \eta_i^2 \mathbb{E}_{\mathbf{J}}[\mathcal{E}_{\mathbf{z}}(\omega_i)],$$

which leads to the desired result by multiplying both sides by 2. The proof is complete. $\qquad\square$

We also need to the following lemma, whose proof can be done by using an induction argument.

**Lemma D.3.** *Let $\{u_t\}_{t=1}^{T}$, $\{A_t\}_{t=1}^{T}$ and $\{B_t\}_{t=1}^{T}$ be three sequences of non-negative numbers such that $u_1 \le A_1$ and*

$$u_t \le A_t + B_t \sup_{i \in [t-1]} u_i, \qquad \forall t \in \{2, 3, \cdots, T\}. \tag{48}$$

*Let $\sup_{t \in [T]} B_t \le B < 1$. Then for all $t \in [T]$,*

$$\sup_{k \in [t]} u_t \le \frac{1}{1-B} \sup_{k \in [t]} A_k. \tag{49}$$

*Proof.* When $t = 1$, (49) holds trivially since $u_1 \le A_1$ and $B < 1$. Now assume for some $t \in \mathbb{N}$ with $2 \le t \le T$,

$$\sup_{i \in [t-1]} u_i \le \frac{1}{1-B} \sup_{i \in [t-1]} A_i.$$

Then, by (48), the above hypothesis, and $B_t \le B$, we have

$$u_t \le A_t + B_t \sup_{i \in [t-1]} u_i \le A_t + \frac{B_t}{1-B} \sup_{i \in [t-1]} A_i \le \sup_{i \in [t]} A_i \left(1 + \frac{B_t}{1-B}\right) \le \sup_{i \in [t]} A_i \frac{1}{1-B}.$$

Consequently,

$$\sup_{k \in [t]} u_t \le \frac{1}{1-B} \sup_{k \in [t]} A_k,$$

thereby showing that indeed (49) holds for $t$. By mathematical induction, (49) holds for every $t \in [T]$. The proof is complete. $\qquad\square$

Now we can bound $\mathbb{E}_{\mathbf{J}}[\mathcal{E}_{\mathbf{z}}(\omega_k)]$ as follows.

**Lemma D.4.** *Let $\eta_1\kappa^2 \leq 1/2$ and for all $t \in [T]$ with $t \geq 2$,*

$$\frac{1}{\eta_t}\sum_{k=1}^{t-1}\frac{1}{k(k+1)}\sum_{i=t-k}^{t-1}\eta_i^2 \leq \frac{1}{4\kappa^2}. \tag{50}$$

*Then for all $t \in [T]$,*

$$\sup_{k\in[t]}\mathbb{E}_{\mathbf{J}}[\mathcal{E}_{\mathbf{z}}(\omega_k)] \leq 8\mathcal{E}_{\mathbf{z}}(0)\sup_{k\in[t]}\left\{\frac{1}{\eta_k k}\sum_{l=1}^{k}\eta_l\right\}. \tag{51}$$

*Proof.* By Lemma D.2, we have (45). Dividing both sides by $\eta_t$, we can relax the inequality as

$$\mathbb{E}_{\mathbf{J}}[\mathcal{E}_{\mathbf{z}}(\omega_t)] \leq 4\mathcal{E}_{\mathbf{z}}(0)\frac{1}{\eta_t t}\sum_{l=1}^{t}\eta_l + 2\kappa^2\frac{1}{\eta_t}\sum_{k=1}^{t-1}\frac{1}{k(k+1)}\sum_{i=t-k}^{t-1}\eta_i^2\sup_{i\in[t-1]}\mathbb{E}_{\mathbf{J}}[\mathcal{E}_{\mathbf{z}}(\omega_i)].$$

In Lemma D.3, we let $u_t = \mathbb{E}_{\mathbf{J}}[\mathcal{E}_{\mathbf{z}}(\omega_t)]$, $A_t = 4\mathcal{E}_{\mathbf{z}}(0)\frac{1}{\eta_t t}\sum_{l=1}^{t}\eta_l$ and

$$B_t = 2\kappa^2\frac{1}{\eta_t}\sum_{k=1}^{t-1}\frac{1}{k(k+1)}\sum_{i=t-k}^{t-1}\eta_i^2.$$

Condition (50) guarantees that $\sup_{t\in[T]}B_t \leq 1/2$. Thus, (49) holds, and the desired result follows by plugging with $B = 1/2$. The proof is complete. $\qquad\square$

Finally, we need the following lemma to bound $\mathcal{E}_{\mathbf{z}}(0)$, whose proof follows from applying the Bernstein Inequality from Lemma A.1.

**Lemma D.5.** *Under Assumption 1, with probability at least $1 - \delta_3$ ($\delta_3 \in{]}0, 1[$), there holds*

$$\mathcal{E}_{\mathbf{z}}(0) \leq Mv + 2Mv\left(\frac{1}{m} + \frac{\sqrt{2}}{\sqrt{m}}\right)\log\frac{2}{\delta_3}.$$

*In particular, if $m \geq 32\log^2\frac{2}{\delta_3}$, then*

$$\mathcal{E}_{\mathbf{z}}(0) \leq 2Mv. \tag{52}$$

*Proof.* Following from (5),

$$\int_Z y^{2l}d\rho \leq \frac{1}{2}l!M^{l-2}\cdot(2M^2v), \qquad \forall l \in \mathbb{N},$$

and

$$\int_Z y^2 d\rho \leq Mv.$$

Therefore,

$$\begin{aligned}
\int_Z |y^2 - \mathbb{E}y^2|^l d\rho &\leq \int_Z \max(|y|^{2l}, (\mathbb{E}y^2)^l)d\rho \\
&\leq \int_Z (|y|^{2l} + (\mathbb{E}y^2)^l)d\rho \\
&\leq \frac{1}{2}l!M^{l-2}\cdot(2M^2v) + (Mv)^l \\
&\leq \frac{1}{2}l!(Mv)^{l-2}(2Mv)^2,
\end{aligned}$$

where for the last inequality we used $v \geq 1$. Applying Lemma A.1, with $\omega_i = y_i^2$ for all $i \in [n]$, $B = Mv$ and $\sigma = 2Mv$, we know that with probability at least $1 - \delta_3$, there holds

$$\frac{1}{n}\sum_{i=1}^{n}y_i^2 - \int_Z y^2 d\rho \leq 2Mv\left(\frac{1}{n} + \frac{2}{\sqrt{n}}\right)\log\frac{2}{\delta_3}.$$

The proof is complete. $\qquad\square$

## D.2 Bounding $\left\|\mathcal{T}^{\frac{1}{2}}\Pi_{k+1}^t(\mathcal{T}_\mathbf{x})\right\|$

**Lemma D.6.** *Assume (39) holds for some $\lambda > 0$ and $\eta_1\kappa^2 \leq 1$. Then*

$$\|\mathcal{T}^{\frac{1}{2}}\Pi_{k+1}^t(\mathcal{T}_\mathbf{x})\|^2 \leq \frac{1}{\sum_{i=k+1}^t \eta_i} + 4\lambda.$$

*Proof.* Note that we have

$$\|\mathcal{T}^{\frac{1}{2}}\Pi_{k+1}^t(\mathcal{T}_\mathbf{x})\| \leq \|\mathcal{T}^{\frac{1}{2}}(\mathcal{T}_\mathbf{x} + \lambda I)^{-\frac{1}{2}}\|\|(\mathcal{T}_\mathbf{x} + \lambda I)^{\frac{1}{2}}\Pi_{k+1}^t(\mathcal{T}_\mathbf{x})\|.$$

Using (39), we can relax the above as

$$\|\mathcal{T}^{\frac{1}{2}}\Pi_{k+1}^t(\mathcal{T}_\mathbf{x})\| \leq 2\|(\mathcal{T}_\mathbf{x} + \lambda I)^{\frac{1}{2}}\Pi_{k+1}^t(\mathcal{T}_\mathbf{x})\|,$$

which leads to

$$\|\mathcal{T}^{\frac{1}{2}}\Pi_{k+1}^t(\mathcal{T}_\mathbf{x})\|^2 \leq 4\|(\mathcal{T}_\mathbf{x} + \lambda I)^{\frac{1}{2}}\Pi_{k+1}^t(\mathcal{T}_\mathbf{x})\|^2.$$

Since

$$\begin{aligned}
\|(\mathcal{T}_\mathbf{x} + \lambda I)^{\frac{1}{2}}\Pi_{k+1}^t(\mathcal{T}_\mathbf{x})\|^2 &= \|(\mathcal{T}_\mathbf{x} + \lambda I)\Pi_{k+1}^t(\mathcal{T}_\mathbf{x})\Pi_{k+1}^t(\mathcal{T}_\mathbf{x})\| \\[2mm]
&\leq \|\mathcal{T}_\mathbf{x}\Pi_{k+1}^t(\mathcal{T}_\mathbf{x})\Pi_{k+1}^t(\mathcal{T}_\mathbf{x})\| + \lambda \\[2mm]
&= \|\mathcal{T}_\mathbf{x}^{\frac{1}{2}}\Pi_{k+1}^t(\mathcal{T}_\mathbf{x})\|^2 + \lambda,
\end{aligned}$$

and with $\eta_t\kappa^2 \leq 1$, $\|\mathcal{T}_\mathbf{x}\| \leq \mathrm{tr}(\mathcal{T}_\mathbf{x}) \leq \kappa^2$, by Lemma B.1,

$$\|\mathcal{T}_\mathbf{x}^{\frac{1}{2}}\Pi_{k+1}^t(\mathcal{T}_\mathbf{x})\|^2 \leq \frac{1}{2\mathrm{e}\sum_{i=k+1}^t \eta_i} \leq \frac{1}{4\sum_{i=k+1}^t \eta_i},$$

we thus derive the desired result. The proof is complete. $\qquad\qquad\square$

## D.3 Deriving Error Bounds

With Lemmas D.4 and D.6, we are ready to estimate the computational variance , $\mathbb{E}_\mathbf{J}\|f_t - g_t\|_\rho^2$, as follows.

**Proposition D.7.** *Assume (39) holds for some $\lambda > 0$, $\eta_1\kappa^2 \leq 1/2$, (50) and (52). Then, we have for all $t \in [T]$,*

$$\mathbb{E}_\mathbf{J}\|\mathcal{S}_\rho\omega_{t+1} - \mathcal{S}_\rho\nu_{t+1}\|_\rho^2 \leq \frac{16Mv\kappa^2}{b}\sup_{k\in[t]}\left\{\frac{1}{\eta_k k}\sum_{l=1}^k \eta_l\right\}\left(\sum_{k=1}^{t-1}\frac{\eta_k^2}{\sum_{i=k+1}^t \eta_i} + 4\lambda\sum_{k=1}^{t-1}\eta_k^2 + \eta_t^2\kappa^2\right). \tag{53}$$

*Proof.* Since $\omega_{t+1}$ and $\nu_{t+1}$ are given by (4) and (19), respectively,

$$\begin{aligned}
\omega_{t+1} - \nu_{t+1} &= (\omega_t - \nu_t) + \eta_t\left\{(\mathcal{T}_\mathbf{x}\nu_t - \mathcal{S}_\mathbf{x}^*\mathbf{y}) - \frac{1}{b}\sum_{i=b(t-1)+1}^{bt}(\langle\omega_t, x_{j_i}\rangle_H - y_{j_i})x_{j_i}\right\} \\[2mm]
&= (I - \eta_t\mathcal{T}_\mathbf{x})(\omega_t - \nu_t) + \frac{\eta_t}{b}\sum_{i=b(t-1)+1}^{bt}\left\{(\mathcal{T}_\mathbf{x}\omega_t - \mathcal{S}_\mathbf{x}^*\mathbf{y}) - (\langle\omega_t, x_{j_i}\rangle_H - y_{j_i})x_{j_i}\right\}.
\end{aligned}$$

Applying this relationship iteratively,

$$\omega_{t+1} - \nu_{t+1} = \Pi_1^t(\mathcal{T}_\mathbf{x})(\omega_1 - \nu_1) + \frac{1}{b}\sum_{k=1}^t\sum_{i=b(k-1)+1}^{bk}\eta_k\Pi_{k+1}^t(\mathcal{T}_\mathbf{x})M_{k,i},$$

where we denote

$$M_{k,i} = (\mathcal{T}_\mathbf{x}\omega_k - \mathcal{S}_\mathbf{x}^*\mathbf{y}) - (\langle\omega_k, x_{j_i}\rangle_H - y_{j_i})x_{j_i}. \tag{54}$$

Introducing with $\omega_1 = \nu_1 = 0$,

$$\omega_{t+1} - \nu_{t+1} = \frac{1}{b}\sum_{k=1}^{t}\sum_{i=b(k-1)+1}^{bk}\eta_k\Pi_{k+1}^t(\mathcal{T}_\mathbf{x})M_{k,i}.$$

Therefore,

$$
\begin{aligned}
\mathbb{E}_\mathbf{J}\|\mathcal{S}_\rho\omega_{t+1} - \mathcal{S}_\rho\nu_{t+1}\|_\rho^2 &= \frac{1}{b^2}\mathbb{E}_\mathbf{J}\left\|\sum_{k=1}^{t}\sum_{i=b(k-1)+1}^{bk}\eta_k\Pi_{k+1}^t(\mathcal{T}_\mathbf{x})M_{k,i}\right\|_\rho^2\\
&= \frac{1}{b^2}\sum_{k=1}^{t}\sum_{i=b(k-1)+1}^{bk}\eta_k^2\mathbb{E}_\mathbf{J}\left\|\Pi_{k+1}^t(\mathcal{T}_\mathbf{x})M_{k,i}\right\|_\rho^2, \quad (55)
\end{aligned}
$$

where for the last equality, we use the fact that if $k \neq k'$, or $k = k'$ but $i \neq i'$[6], then

$$\mathbb{E}_\mathbf{J}\langle\Pi_{k+1}^t(\mathcal{T}_\mathbf{x})M_{k,i}, \Pi_{k'+1}^t(\mathcal{T}_\mathbf{x})M_{k',i'}\rangle_\rho = 0.$$

Indeed, if $k \neq k'$, without loss of generality, we consider the case $k < k'$. Recalling that $M_{k,i}$ is given by (54) and that given any $\mathbf{z}$, $f_k$ is depending only on $\mathbf{J}_1, \cdots, \mathbf{J}_{k-1}$, we thus have

$$
\begin{aligned}
&\mathbb{E}_\mathbf{J}\langle\Pi_{k+1}^t(\mathcal{T}_\mathbf{x})M_{k,i}, \Pi_{k'+1}^t(\mathcal{T}_\mathbf{x})M_{k',i'}\rangle_\rho\\
&= \mathbb{E}_{\mathbf{J}_1,\cdots,\mathbf{J}_{k'-1}}\langle\Pi_{k+1}^t(\mathcal{T}_\mathbf{x})M_{k,i}, \Pi_{l+1}^t(\mathcal{T}_\mathbf{x})\mathbb{E}_{\mathbf{J}_{k'}}[M_{k',i'}]\rangle_\rho = 0.
\end{aligned}
$$

If $k = k'$ but $i \neq i'$, without loss of generality, we assume $i < i'$. By noting that $\omega_k$ is depending only on $\mathbf{J}_1, \cdots, \mathbf{J}_{k-1}$ and $M_{k,i}$ is depending only on $\omega_k$ and $z_{j_i}$ (given any sample $\mathbf{z}$),

$$
\begin{aligned}
&\mathbb{E}_\mathbf{J}\langle\Pi_{k+1}^t(\mathcal{T}_\mathbf{x})M_{k,i}, \Pi_{k+1}^t(\mathcal{T}_\mathbf{x})M_{k,i'}\rangle_\rho\\
&= \mathbb{E}_{\mathbf{J}_1,\cdots,\mathbf{J}_{k-1}}\langle\Pi_{k+1}^t(\mathcal{T}_\mathbf{x})\mathbb{E}_{j_i}[M_{k,i}], \Pi_{l+1}^t(\mathcal{T}_\mathbf{x})\mathbb{E}_{j_{i'}}[M_{k,i'}]\rangle_\rho = 0.
\end{aligned}
$$

Using the isometry property (17) to (55),

$$\mathbb{E}_\mathbf{J}\left\|\Pi_{k+1}^t(\mathcal{T}_\mathbf{x})M_{k,i}\right\|_\rho^2 = \mathbb{E}_\mathbf{J}\left\|\mathcal{T}^{\frac{1}{2}}\Pi_{k+1}^t(\mathcal{T}_\mathbf{x})M_{k,i}\right\|_H^2 \leq \left\|\mathcal{T}^{\frac{1}{2}}\Pi_{k+1}^t(\mathcal{T}_\mathbf{x})\right\|^2\mathbb{E}_\mathbf{J}\left\|M_{k,i}\right\|_H^2,$$

and by applying the inequality $\mathbb{E}[\|\xi - \mathbb{E}[\xi]\|_H^2] \leq \mathbb{E}[\|\xi\|_H^2]$,

$$\mathbb{E}_\mathbf{J}\|M_{k,i}\|_H^2 \leq \mathbb{E}_\mathbf{J}\|(\langle\omega_k, x_{j_i}\rangle_H - y_{j_i})x_{j_i}\|_H^2 \leq \kappa^2\mathbb{E}_\mathbf{J}[(\langle\omega_k, x_{j_i}\rangle_H - y_{j_i})^2] = \kappa^2\mathbb{E}_\mathbf{J}[\mathcal{E}_\mathbf{z}(\omega_k)],$$

where for the last inequality we use (3). Therefore,

$$\mathbb{E}_\mathbf{J}\|\mathcal{S}_\rho\omega_{t+1} - \mathcal{S}_\rho\nu_{t+1}\|_\rho^2 \leq \frac{\kappa^2}{b}\sum_{k=1}^{t}\eta_k^2\left\|\mathcal{T}^{\frac{1}{2}}\Pi_{k+1}^t(\mathcal{T}_\mathbf{x})\right\|^2\mathbb{E}_\mathbf{J}[\mathcal{E}_\mathbf{z}(\omega_k)].$$

According to Lemma D.4, we have (51). It thus follows that

$$\mathbb{E}_\mathbf{J}\|\mathcal{S}_\rho\omega_{t+1} - \mathcal{S}_\rho\nu_{t+1}\|_\rho^2 \leq \frac{8\mathcal{E}_\mathbf{z}(0)\kappa^2}{b}\sup_{k\in[t]}\left\{\frac{1}{\eta_k k}\sum_{l=1}^{k}\eta_l\right\}\sum_{k=1}^{t}\eta_k^2\left\|\mathcal{T}^{\frac{1}{2}}\Pi_{k+1}^t(\mathcal{T}_\mathbf{x})\right\|^2.$$

Now the proof can be finished by applying Lemma D.6 which tells us that

$$
\begin{aligned}
\sum_{k=1}^{t}\eta_k^2\left\|\mathcal{T}^{\frac{1}{2}}\Pi_{k+1}^t(\mathcal{T}_\mathbf{x})\right\|^2 &= \sum_{k=1}^{t-1}\eta_k^2\left\|\mathcal{T}^{\frac{1}{2}}\Pi_{k+1}^t(\mathcal{T}_\mathbf{x})\right\|^2 + \eta_t^2\left\|\mathcal{T}^{\frac{1}{2}}\right\|^2\\
&\leq \sum_{k=1}^{t-1}\frac{\eta_k^2}{\sum_{i=k+1}^{t}\eta_i} + 4\lambda\sum_{k=1}^{t-1}\eta_k^2 + \eta_t^2\kappa^2,
\end{aligned}
$$

and (52) to the above. The proof is complete. $\qquad\square$

Setting $\eta_t = \eta_1 t^{-\theta}$ for some appropriate $\eta_1$ and $\theta$ in the above proposition, we get the following explicitly upper bounds for $\mathbb{E}_{\mathbf{J}}\|\mathcal{S}_\rho\omega_t - \mathcal{S}_\rho\omega_t\|_\rho^2$.

**Proposition D.8.** *Assume (39) holds for some $\lambda > 0$ and (52). Let $\eta_t = \eta_1 t^{-\theta}$ for all $t \in [T]$, with $\theta \in [0,1[$ and*

$$0 < \eta_1 \le \frac{t^{\min(\theta, 1-\theta)}}{8\kappa^2(\log t + 1)}, \qquad \forall t \in [T]. \tag{56}$$

*Then, for all $t \in [T]$,*

$$\mathbb{E}_{\mathbf{J}}\|\omega_{t+1} - \nu_{t+1}\|_\rho^2 \le \frac{16Mv\kappa^2}{b(1-\theta)}\left(5\eta_1 t^{-\min(\theta,1-\theta)} + 8\lambda\eta_1^2 t^{(1-2\theta)_+}\right)(1 \vee \log t). \tag{57}$$

*Proof.* We will use Proposition D.7 to prove the result. Thus, we need to verify the condition (50). Note that

$$\sum_{k=1}^{t-1}\frac{1}{k(k+1)}\sum_{i=t-k}^{t-1}\eta_i^2 = \sum_{i=1}^{t-1}\eta_i^2\sum_{k=t-i}^{t-1}\frac{1}{k(k+1)} = \sum_{i=1}^{t-1}\eta_i^2\left(\frac{1}{t-i} - \frac{1}{t}\right) \le \sum_{i=1}^{t-1}\frac{\eta_i^2}{t-i}.$$

Substituting with $\eta_i = \eta i^{-\theta}$, and by Lemma A.4,

$$\sum_{k=1}^{t-1}\frac{1}{k(k+1)}\sum_{i=t-k}^{t-1}\eta_i^2 \le \eta_1^2\sum_{i=1}^{t-1}\frac{i^{-2\theta}}{t-i} \le 2\eta_1^2 t^{-\min(2\theta,1)}(\log t + 1).$$

Dividing both sides by $\eta_t$ $(= \eta t^{-\theta})$, and then using (56),

$$\frac{1}{\eta_t}\sum_{k=1}^{t-1}\frac{1}{k(k+1)}\sum_{i=t-k}^{t-1}\eta_i^2 \le 2\eta_1 t^{-\min(\theta,1-\theta)}(\log t + 1) \le \frac{1}{4\kappa^2}.$$

This verifies (50). Note also that by taking $t = 1$ in (56), for all $t \in [T]$,

$$\eta_t\kappa^2 \le \eta_1\kappa^2 \le \frac{1}{8\kappa^2} \le \frac{1}{2}.$$

We thus can apply Proposition D.7 to derive (53). What remains is to control the right hand side of (53). Since

$$\sum_{k=1}^{t-1}\frac{\eta_k^2}{\sum_{i=k+1}^{t}\eta_i} = \eta_1\sum_{k=1}^{t-1}\frac{k^{-2\theta}}{\sum_{i=k+1}^{t}i^{-\theta}} \le \eta_1\sum_{k=1}^{t-1}\frac{k^{-2\theta}}{(t-k)t^{-\theta}},$$

combining with Lemma A.4,

$$\sum_{k=1}^{t-1}\frac{\eta_k^2}{\sum_{i=k+1}^{t}\eta_i} \le 2\eta_1 t^{-\min(\theta,1-\theta)}(\log t + 1).$$

Also, by Lemma A.2,

$$\frac{1}{\eta_k k}\sum_{l=1}^{k}\eta_l = \frac{1}{k^{1-\theta}}\sum_{l=1}^{k}l^{-\theta} \le \frac{1}{1-\theta},$$

and by Lemma A.3,

$$\sum_{k=1}^{t-1}\eta_k^2 = \eta_1^2\sum_{k=1}^{t-1}k^{-2\theta} \le \eta_1^2 t^{\max(1-2\theta,0)}(\log t + 1).$$

Introducing the last three estimates into (53) and using that $\eta_t^2\kappa^2 \le \eta_1 t^{-\theta}$ by (56), we get the desired result. The proof is complete. $\qquad\square$

Collect some of the above analysis, we get the following result for the computational variance.

**Theorem D.9.** *Under Assumptions 1 and 3, let $\delta_2 \in ]0,1[$, $\frac{9\kappa^2}{m}\log\frac{m}{\delta_2} \le \lambda \le \|\mathcal{T}\|$, $\delta_3 \in ]0,1[$, $m \ge 32\log^2\frac{2}{\delta_3}$, and $\eta_t = \eta t^{-\theta}$ for all $t \in [T]$, with $\theta \in [0,1[$ and $\eta$ such that (56). Then, with probability at least $1 - \delta_2 - \delta_3$, (57) holds for all $t \in [T]$.*

# E  Deriving Total Error Bounds

The purpose of this section is to derive total error bounds.

## E.1  Attainable Case

We have the following general theorem for $\zeta \geq 1/2$, with which we prove our main results stated in Section 3.

**Theorem E.1.** *Under Assumptions 1, 2 and 3, let $\zeta \geq 1/2$, $T \in \mathbb{N}$ with $T \geq 3$, $\delta \in ]0, 1[$, $\eta_t = \eta \kappa^{-2} t^{-\theta}$ for all $t \in [T]$, with $\theta \in [0, 1[$ and $\eta$ such that*

$$0 < \eta \leq \frac{t^{\min(\theta, 1-\theta)}}{8(\log t + 1)}, \qquad \forall t \in [T]. \tag{58}$$

*If for some $\epsilon \in ]0, 1]$,*

$$m \geq \left( \frac{18\kappa^2}{\epsilon \|\mathcal{T}\|} \log \left( \frac{27\kappa^2}{\epsilon \|\mathcal{T}\| \delta} \right) \right)^{1/\epsilon}, \tag{59}$$

*then the following holds with probability at least $1 - \delta$: for all $t \in [T]$,*

$$\mathbb{E}_{\mathbf{J}}[\mathcal{E}(\omega_{t+1})] - \inf_{\omega \in H} \mathcal{E}(\omega) \leq q_1 (\eta t^{1-\theta})^{-2\zeta} + q_2 m^{\gamma(1-\epsilon)-1}(1 \vee \eta^2 m^{2\epsilon-2} t^{2-2\theta})(\log T)^2 \log^2 \frac{12}{\delta}$$
$$+ q_3 \eta b^{-1}(t^{-\min(\theta, 1-\theta)} \vee m^{\epsilon-1} \eta t^{(1-2\theta)_+}) \log T. \tag{60}$$

*Here, $q_1 = 2R^2 \zeta^{2\zeta}$, $q_2 = \frac{800(R\kappa^{2\zeta} + \sqrt{M})^2 (\kappa/\sqrt{\|\mathcal{T}\|} + \sqrt{2\sqrt{v}c_\gamma/\|\mathcal{T}\|^\gamma})^2}{(1-\theta)^2}$, and $q_3 = \frac{208Mv}{1-\theta}$.*

*Proof.* Let $\lambda = \|\mathcal{T}\| m^{\epsilon-1}$. Clearly, $\lambda \leq \|\mathcal{T}\|$. For any $A \geq 0$ and $B \geq 1$, by applying (25) with $\zeta = 1$, $x = (Bm)^\epsilon$ and $c = \frac{\epsilon}{2AB^\epsilon}$,

$$A \log(Bm) = \frac{A}{\epsilon} \log((Bm)^\epsilon) \leq \frac{A}{\epsilon} \log \left( \frac{2AB^\epsilon}{e\epsilon} \right) + \frac{1}{2} m^\epsilon \leq \frac{A}{\epsilon} \log \left( \frac{AB}{\epsilon} \right) + \frac{1}{2} m^\epsilon. \tag{61}$$

Using the above inequality with $A = \frac{9\kappa^2}{\|\mathcal{T}\|}$ and $B = \frac{1}{\delta_2}$, one can prove that the condition (59) ensures that $\frac{9\kappa^2}{m} \log \frac{m}{\delta_2} \leq \lambda$ is satisfied with $\delta_2 = \frac{\delta}{3}$, Therefore, by Lemma C.3, (39) holds with probability at least $1 - \delta_2$. Similarly the condition (59) implies that $m \geq 32 \log^2 \frac{2}{\delta_3}$ is satisfied with $\delta_3 = \frac{\delta}{3}$, and thus by Lemma D.5, (52) holds with probability at least $1 - \delta_3$. Combining with Lemma C.2, by taking the union bound, we know that with probability at least $1 - \delta_1 - \delta_2 - \delta_3$, (39), (52) and (34) hold for all $k \in [T]$. Now, we can apply Propositions C.5 and D.8 to get (42) and (57). Noting that by (56), $\sqrt{2}\eta \leq 1$, and by a simple calculation, we derive from (42) that

$$\|\mathcal{S}_\rho \nu_{t+1} - \mathcal{S}_\rho \mu_{t+1}\|_\rho^2$$
$$\leq \frac{400(R\kappa^{2\zeta} + \sqrt{M})^2 (\kappa/\sqrt{\|\mathcal{T}\|} + \sqrt{2\sqrt{v}c_\gamma/\|\mathcal{T}\|^\gamma})^2}{(1-\theta)^2} m^{\gamma(1-\epsilon)-1}(1 \vee \lambda^2 \eta^2 \kappa^{-4} t^{2-2\theta} \vee \log^2 t) \log^2 \frac{4}{\delta_1}$$
$$\leq \frac{400(R\kappa^{2\zeta} + \sqrt{M})^2 (\kappa/\sqrt{\|\mathcal{T}\|} + \sqrt{2\sqrt{v}c_\gamma/\|\mathcal{T}\|^\gamma})^2}{(1-\theta)^2} m^{\gamma(1-\epsilon)-1}(1 \vee \eta^2 m^{2\epsilon-2} t^{2-2\theta})(\log T)^2 \log^2 \frac{4}{\delta_1},$$

where for the last inequality, we used $\|\mathcal{T}\| \leq \kappa^2$. Similarly, by a simple calculation, we get from (57) that

$$\mathbb{E}_{\mathbf{J}}\|\mathcal{S}_\rho \omega_{t+1} - \mathcal{S}_\rho \nu_{t+1}\|_\rho^2 \leq \frac{208Mv}{b(1-\theta)}(\eta t^{-\min(\theta, 1-\theta)} \vee \lambda \eta^2 \kappa^{-2} t^{(1-2\theta)_+})(1 \vee \log t)$$
$$\leq \frac{208Mv}{b(1-\theta)}(\eta t^{-\min(\theta, 1-\theta)} \vee m^{\epsilon-1} \eta^2 t^{(1-2\theta)_+}) \log T.$$

Letting $\delta_1 = \frac{\delta}{3}$, and introducing the above estimates and (27) into (16), we get (60). The proof is complete. $\qquad \square$

*Proof of Theorem 3.3.* By choosing $\epsilon = 1 - \frac{1}{2\zeta+\gamma}$ and $\theta = 0$ in Theorem E.1, then the condition (59) reduces to $m \geq m_\delta$, where

$$m_\delta = \left( \frac{18\kappa^2 p}{\|\mathcal{T}\|} \log\left( \frac{27\kappa^2 p}{\|\mathcal{T}\|\delta} \right) \right)^p, \quad p = \frac{2\zeta+\gamma}{2\zeta+\gamma-1}. \tag{62}$$

The desired result thus follows by applying Theorem E.1. $\qquad\square$

### E.2 Non Attainable Case

**Theorem E.2.** *Under Assumptions 1, 2 and 3, let $\zeta \leq 1/2$, $T \in \mathbb{N}$ with $T \geq 3$, $\delta \in ]0,1[$, $\eta_t = \eta\kappa^{-2}t^{-\theta}$ for all $t \in [T]$, with $\theta \in [0,1[$ and $\eta$ such that (58) and for some $\epsilon \in ]0,1]$, (59) holds. Then the following holds with probability at least $1-\delta$: for all $t \in [T]$,*

$$\mathbb{E}_{\mathbf{J}}[\mathcal{E}(\omega_{t+1})] - \inf_{\omega \in H} \mathcal{E}(\omega) \lesssim (\eta t^{1-\theta})^{-2\zeta} + m^{\gamma(1-\epsilon)-1}(1 \vee \eta^2 m^{2\epsilon-2}t^{2-2\theta}) \left(1 \vee \eta t^{1-\theta}\right)^{1-2\zeta} \log^2 t \log^2 \frac{4}{\delta_1}$$
$$+ \eta b^{-1}(t^{-\min(\theta,1-\theta)} \vee m^{\epsilon-1}\eta t^{(1-2\theta)+}) \log T. \tag{63}$$

*Proof.* The proof is similar to that for Theorem E.1. We include the sketch only. Similar to the proof of Theorem E.1, one can prove that with probability at least $1 - \delta_1 - \delta_2 - \delta_3$, (39), (52) and (35) hold for all $k \in [T]$. Now, we can apply Propositions C.5 and D.8 to get (43) and (57). Noting that by (56), $\sqrt{2}\eta \leq 1$, and by a simple calculation, we derive from (43) that

$$\|\mathcal{S}_\rho \nu_{t+1} - \mathcal{S}_\rho \mu_{t+1}\|_\rho^2 \leq \frac{400 \left( \kappa^{2\zeta}\left(1 \vee \frac{2\eta t^{1-\theta}}{1-\theta}\right)^{\frac{1}{2}-\zeta} + \sqrt{M} \right)^2 (\kappa/\sqrt{\|\mathcal{T}\|} + \sqrt{2\sqrt{v}c_\gamma/\|\mathcal{T}\|^\gamma})^2}{(1-\theta)^2}$$
$$\times m^{\gamma(1-\epsilon)-1}(1 \vee \lambda^2\eta^2\kappa^{-4}t^{2-2\theta} \vee \log^2 t) \log^2 \frac{4}{\delta_1}.$$

The rest of the proof parallelizes to that for Theorem E.1. $\qquad\square$

**Remark E.3.** *Letting $\theta = 0$ in the above theorem, and ignoring the logarithmic terms, the bound (63) reads as*

$$\mathbb{E}_{\mathbf{J}}[\mathcal{E}(\omega_{t+1})] - \inf_{\omega \in H} \mathcal{E}(\omega) \lesssim (\eta t)^{-2\zeta} + m^{\gamma(1-\epsilon)-1}(1 \vee m^{\epsilon-1}\eta t)^2 \left(1 \vee \eta t\right)^{1-2\zeta} + \eta b^{-1}(1 \vee m^{\epsilon-1}\eta t).$$

**Remark E.4.** *Better bounds for the case $\zeta \leq 1/2$ will be proved in the longer version of this paper.*

## Footnotes

[6]This is possible only when $b \geq 2$.