[Reviews · NeurIPS 2016]

Reviewer 1

Summary

A generalization error analysis for the stochastic optimization method on an RKHS is given. The error bound is characterized by the smoothness of the true function and the capacity of the model. Moreover, conditions of the stepsize, the minibatch size, and the number of steps are also given. Those theoretical analysis is interesting although it is a bit incremental.

Qualitative Assessment

This is a theoretical paper that gives the convergence rate of the stochastic gradient method (SGM) for an optimization problem on an RKHS. The analysis captures more information about the capacity of the model so that it gives a sharper bound than the existing ones. The convergence analysis is given in terms of the mini-batch size, the sample sizes, and the step size and the bound is characterized by the capacity of the model and the "smoothness" of the true function. It can be shown that an early stopping gives the minimax optimal generalization error. Significance: The paper deals with an important problem: the empirical risk minimization problem over an RKHS which is interested in a wide range of the NIPS audiences. The derived learning rate improves the existing bound. It is significant. Novelty: It is natural to consider the capacity condition to derive a fast learning rate as discussed in the paper [13]. The current paper gives such analysis in the online setting. The analysis is interesting. On the other hand, I feel that the contribution is rather incremental to the work [11] (ICML2016). I think more discussions about the comparison with [11] would be helpful for the readers. Clarity: The paper is clearly written.

Confidence in this Review

2-Confident (read it all; understood it all reasonably well)


Reviewer 2

Summary

This paper studies multi-pass stochastic gradient methods (SGM) applied to the least squares problem in a general Hilbert space setting. The authors show that optimal learning rates can be achieved by tuning the number of passes under different choices of step-sizes and mini-batch sizes, in the spirit that the number of passes acts as a regularization parameter. The paper is clearly written. The paper looks novel and interesting.

Qualitative Assessment

The paper generalizes and refines the paper "learning with incremental iterative regularization" in NIPS 2015 by considering different step-sizes, mini-batch strategy, stochastic choices of gradients and capacity-dependent learning rates. I only have minor comments here: line 227: "does not necessarily belongs to" should be "does not necessarily belong to" proof of Lemma B.3: "The case for \xi\geq1/2 is similar to that" should be "The case for \xi < 1/2"? Also note that the case \xi=1/2 is covered both in (S14) and (S15).

Confidence in this Review

2-Confident (read it all; understood it all reasonably well)


Reviewer 3

Summary

This paper establishes optimal rates (up to log factors here and throughout this review) using multi-pass mini-batch SGM. Most results hold with moderately high probability (failure probability $1 - 1/m$), and it appears that all the guarantees are valid only for $m$ ``large enough'' (greater than $m_0$), which might not be known without additional information related to the capacity. In the case of b = 1 (non-mini-batch), optimal rates are obtained by a method which depends on the capacity only through the stopping time (Corollary 3.3). Corollary 3.5 appears to be suboptimal; it is similar to Corollary 3.5 but involves taking $b = \sqrt{m}$. Corollaries 3.6 and 3.8 have stopping times ostensibly independent of the capacity, but either the learning rate (Corollary 3.6) or the batch size (Corollary 3.8) depend on the capacity. Corollary 3.9 is similar to Corollary 3.3 in the sense of providing a guarantee for a method that depends on the capacity only through the stopping time, but this later result now uses $b = m$, corresponding to a batch-like multi-pass SGM method. Some experimental results exhibit the expected behavior from the bias and sample error.

Qualitative Assessment

This work provides a strong contribution in that it apparently is the first work to net optimal rates (up to log factors) for SGM, and moreover, it also handles a mini-batch analysis which includes the (full) batch method as a special case. Such rates previously had been established only for the (batch) ridge regression method. My interpretation of what all the results actually show is given in the Summary. I find the current solution of relying on cross-validation for adaptation to be a bit of an inelegant cop-out (even if there is a theoretically-supported method for using it); given that several of your corollaries provide a guarantee where $\zeta$ and $\gamma$ enter the picture only through $T^*$, can you provide a self-monitoring method that decides when to stop? In particular, I find the most exciting results to be Corollaries 3.3 and 3.9, as only the stopping time depends on the (unknown) capacity parameters, and so such an online stopping mechanism might be possible. An issue with all of the results is that they are only valid for ``large enough'' $m$, which cannot be verified without knowing the unknown capacity parameters. I am somewhat willing to overlook this weakness, however, as it may be unavoidable. It would have been useful if the authors briefly sketched, in the main paper, how they proved bounds on each of the three components in their error decomposition. Even a few sentences for each would suffice (given the space constraints, much more may be tricky). As I explain below, this will help both with the clarity of the technical portion and the impact of this work. I did not actually get much out of the experimental results near the end, so I would argue for partially cutting them in favor of giving readers more information about the technical novelty of your results and particular sub-results that might be reused in other works. This could help boost the impact of your work. At a minimum, the current way the Appendix reads is somewhat adversarial. For starters, the appendix is not searchable. It appears that it is just a sequence of images, which is rather annoying. Using 'pdfjam' in the future should lead to a proper splitting of PDFs without sacrificing the text-encoding. But most importantly, if you provide proof sketches in the appendix, then readers will have some idea of the overall argument, and this will help serve as a guide to the proofs. This will help with technical reuse so that other works can continue in this line. In its current state, the appendix is more or less a barrage of proofs without much indication of the overall strategy (I mean this constructively!). I therefore found the proofs very hard to check (although I read some of the initial proofs and they appear to be correct), and this certainly correlates to the results being hard to adapt.

Confidence in this Review

2-Confident (read it all; understood it all reasonably well)


Reviewer 4

Summary

This paper analyzes the generalization performance of stochastic gradient descent in a multi-pass setting, focusing in particular on the case of stochastic gradient descent as applied to non-parametric regression, and analyzing mini-batches in addition to the multiple passes. This is an interesting setting to analyze because, whereas most practitioners use multiple passes of SGD for training, much of the most fundamental theory surrounding SGD only holds in the one-pass setting.

Qualitative Assessment

This paper works towards remedying an important hole in our current understanding of stochastic gradient descent --- namely, that while most theory applies to 1-pass SGD, most practitioners find best results when doing multi-pass SGD. The results all look reasonable, although I didn't check the proofs. I liked Section 3.4 and would suggest making it a bit more prominent to readers (at least indicating that in the intro that this is where the proof idea is described). From my perspective, the main drawback of the paper was that it was presented in such an abstract setting (non-parametric regression on Hilbert spaces) that it was hard to understand the implications for vanilla SGD. Since the main motivation for this problem is remedying a gap between theory and practice, I think this drawback is substantial. That being said, perhaps others will later distill these results to be more understandable to practitioners, so I am currently in favor of accepting the paper, though I won't necessarily push for it. Some questions for the authors that I'd like to see answered in the rebuttal (i.e., these are my most important questions): -Can you explain the implications of your results in the typical SGD setting (d-dimensional vector space, \ell^2-norm bounds on parameters and gradients)? In particular, what are the differences from previous work in this setting? -Can you be more precise about what you mean by "same rate as ridge regression"? Does this mean the same dependence on m, or also same dependence on other parameters (radius R, etc.)? Other questions (less important): -Can you say anything about SGD in the setting where we just pass over the data from left to right?

Confidence in this Review

2-Confident (read it all; understood it all reasonably well)